# Coordinated crosstalk between microtubules and actin by a spectraplakin regulates lumen formation and branching

**Delia Ricolo[1,2], Sofia J Araujo[1,2]***

[1]Department of Genetics, Microbiology and Statistics, School of Biology, University of Barcelona, Barcelona, Spain; [2]Institute of Biomedicine University of Barcelona (IBUB), Barcelona, Spain

**Abstract** Subcellular lumen formation by single-cells involves complex cytoskeletal remodelling. We have previously shown that centrosomes are key players in the initiation of subcellular lumen formation in *Drosophila melanogaster*, but not much is known on the what leads to the growth of these subcellular luminal branches or makes them progress through a particular trajectory within the cytoplasm. Here, we have identified that the spectraplakin Short-stop (Shot) promotes the crosstalk between MTs and actin, which leads to the extension and guidance of the subcellular lumen within the tracheal terminal cell (TC) cytoplasm. Shot is enriched in cells undergoing the initial steps of subcellular branching as a direct response to FGF signalling. An excess of Shot induces ectopic acentrosomal luminal branching points in the embryonic and larval tracheal TC leading to cells with extra-subcellular lumina. These data provide the first evidence for a role for spectraplakins in single-cell lumen formation and branching.

**\*For correspondence:**
sofiajaraujo@ub.edu

**Competing interests:** The authors declare that no competing interests exist.

## Introduction

Cell shape is intrinsically connected with cell function and varies tremendously throughout nature. Tissue and organ morphogenesis rely on cellular branching mechanisms that can be multicellular or arise within a single-cell. Through extensive cellular remodelling, this so-called single-cell or subcellular branching, transforms an initially relatively symmetrical unbranched cell into an elaborate branched structure. These cellular remodelling events are triggered by widespread cytoskeletal changes and cell membrane growth, which allow these branched cells to span very large areas and accomplish their final function. Despite this clear link between morphology and function, not much is known about the signalling events that trigger the formation of these subcellular branches or what makes them choose a particular trajectory within the cytoplasm of the cell.

In *Drosophila melanogaster*, tracheal system terminal cells (TCs) and nervous system dendrites are models for these subcellular branching processes. During tracheal embryonic through larval development, the generation of single-cell branched structures by TCs is characterised by extensive remodelling of the MT network and actin cytoskeleton, followed by vesicular transport and membrane dynamics (*Best, 2019*; *Gervais and Casanova, 2010*; *Sigurbjörnsdóttir et al., 2014*). During embryonic development, TCs, as tip-cells, lead multicellular branch migration and extension in response to Bnl-Btl signalling, which induces the expression of *Drosophila* serum response factor (DSRF/*blistered (bs)*) and its downstream effectors (*Affolter et al., 1994*; *Posern and Treisman, 2006*). Although epithelial in origin, TCs do not have a canonical apical-basal polarity, and, as they migrate, extend numerous filopodia on their basolateral membrane, generating transient protrusive branches at the leading edge (*Lebreton and Casanova, 2014*). As a consequence, they display a polarity similar to that of a migrating mesenchymal cell (*Fischer et al., 2019*).

While migrating and elongating, the TC invaginates a subcellular tube from its apical membrane, at the contact site with the stalk cell (*Gervais and Casanova, 2010*). The generation of this de novo subcellular lumen can be considered the beginning of the single-cell branching morphogenesis of this cell, which continues throughout larval stages to generate an elaborate single-cell branched structure with many subcellular lumina (*Best, 2019*).

We have previously shown that centrosomes are key players in the initiation of subcellular branching events during embryogenesis. Here, they act as microtubule organising centres (MTOCs) mediating the formation of single or multiple-branched structures depending on their numbers in the TC (*Ricolo et al., 2016*). Centrosomes organise the growth of MT-bundles toward the elongating basolateral edge of the TC. These MTs have been suggested to serve both as trafficking mediators, guiding vesicles for delivery of membrane material, and as mechanical and structural stabilisers for the new subcellular lumen (*Best, 2019*). Actin filaments are present at the growing tip, the basolateral and the luminal membrane of the TC, and actin-regulating factors such as DSRF, Enabled (Ena) and Moesin (Moe) have been shown to contribute to TC morphogenesis (*Gervais and Casanova, 2010*; *Guillemin et al., 1996*; *Schottenfeld-Roames et al., 2014*). During TC elongation, the lumen extends along with the cell, stabilizing the elongating cell body and maintaining a more or less constant distance between its own tip and the migrating tip of the cell (*Gervais and Casanova, 2010*). At the TC basolateral side, a dynamic actin pool integrates the filopodia and aligns the growing subcellular tube with the elongation axis (*JayaNandanan et al., 2014*; *Okenve-Ramos and Llimargas, 2014*; *Oshima et al., 2006*). Together, MT-bundles and the basolateral actin pool are necessary for subcellular lumen formation (*Gervais and Casanova, 2010*). However, not much is known on how these two cytoskeletal structures are coordinated within the TC.

By the time the larva hatches, TCs have elongated and grown a full-length lumen, which becomes gas-filled along with the rest of the tracheal system. In the larva, terminal cells ramify extensively and form many new cytoplasmatic extensions each with a membrane-bound lumen creating tiny subcellular tubes that supply the targets with oxygen (*Baer et al., 2007*; *Ghabrial et al., 2011*; *Whitten, 1957*). At larval stages, sprouting and extension of new branches in response to local hypoxia is generally considered to occur by essentially the same molecular mechanisms as the initial tube invagination and cell extension in the embryo (*Jarecki et al., 1999*; *Sigurbjörnsdóttir et al., 2014*). However, not much is known about how hypoxic signalling is transduced into cytoskeletal modulation to achieve the single-cell branching morphogenesis of the TC. Also, what coordinates the crosstalk between microtubules and actin at the basolateral growing tip, how cell elongation is stabilised by lumen formation and how both processes remain coordinated is still poorly understood in both embryonic and larval TCs.

Spectraplakins are giant conserved cytoskeletal proteins with a complex multidomain architecture capable of binding MTs and actin. They have been reported to crosslink MT minus-ends to actin-networks, making MT-bundles more stable and resistant to catastrophe (*Dogterom and Koenderink, 2019*). Loss of spectraplakins has been shown in vivo to have remarkable effects on microtubule organisation, cell polarity, cell morphology, and cell adhesion (*Röper et al., 2002*; *Suozzi et al., 2012*). *Drosophila* has a single spectraplakin, encoded by *short-stop* (*shot*) (*Gregory and Brown, 1998*; *Lee et al., 2000*; *Röper et al., 2002*). *shot* mutants display pleiotropic phenotypes in wing adhesion, axon and dendrite outgrowth, tracheal fusion, muscle-tendon junction, dorsal closure, oocyte specification and patterning, photoreceptor polarity and perinuclear microtubule network formation (*Gregory and Brown, 1998*; *Khanal et al., 2016*; *Lee and Kolodziej, 2002a*, *Lee and Kolodziej, 2002b*; *Mui et al., 2011*; *Subramanian et al., 2003*; *Sun et al., 2019*). Shot has been shown to bind both the microtubule plus-end-binding EB1 and the microtubule minus-end-binding protein Patronin, required for the establishment of acentrosomal microtubule networks (*Khanal et al., 2016*; *Nashchekin et al., 2016*; *Subramanian et al., 2003*). It also has been shown to bind actin and to crosslink MTs and actin contributing to cytoskeletal organisation and dynamics (*Applewhite et al., 2010*; *Booth et al., 2014*; *Lee and Kolodziej, 2002b*).

In the present study, we uncover a novel role for the spectraplakin Shot in subcellular lumen formation and branching. Our results show that *shot* loss-of-function (LOF) leads to cells deficient in de novo subcellular lumen formation at embryonic stages. We show that Shot promotes the crosstalk between microtubules and actin, which leads to the extension and guidance of the subcellular lumen within the TC cytoplasm. We observe that Shot levels are enriched in cells undergoing the initial steps of subcellular branching as a direct response to FGF signalling. And an excess of Shot induces

ectopic acentrosomal branching points in the embryonic and larval tracheal TC leading to cells with extra-subcellular lumina. Furthermore, we find that Tau protein can functionally replace Shot in subcellular lumen formation and branching.

## Results

### Loss of shot causes defects in de novo subcellular lumen formation

Shot is expressed during *Drosophila* development in several tissues such as the epidermis, the midgut primordia, the trachea and the nervous system (*Lee and Kolodziej, 2002a*; *Röper and Brown, 2003*). We began by analysing the effect of *shot* LOF during TC subcellular lumen formation. To do so, we analysed dorsal (DB) and ganglionic branch (GB) TCs at late stages of embryogenesis (st.15/16) (*Figure 1A,B*).

The *shot*[3] null mutant TC phenotype consisted in subcellular lumen elongation defects with a penetrance of 100% per embryo (n = 40) and 62.5% per TC (n = 400) (*Figure 1C,D and F–I and E*). Of the total mutant TCs analysed, 25% did not develop a subcellular lumen (n = 600, *Figure 1L*). This phenotype resembled the previously reported for *blistered* (*bs*) mutants (*Guillemin et al., 1996*). *bs* encodes the transcription factor DSRF that regulates TC fate induction in response to Branchless-Breathless (Bnl-Btl) signalling (*Gervais and Casanova, 2011*; *Guillemin et al., 1996*). However, we observed that DSRF was properly accumulated in *shot*[3] TC nuclei (*Figure 1D*), discarding a possible effect of Shot in TC fate induction.

To analyse if the *shot* phenotype was tissue autonomous, we expressed *shot*-RNAi to knockdown Shot in all tracheal cells and found that, like in null mutant conditions, 60% of TCs analysed (n = 300) at the tip of the DBs (n = 150) or GBs (n = 150) were affected in subcellular lumen formation (*Figure 1E and J,K*). Of these, 20% did not develop a terminal lumen at all (*Figure 1L*).

*shot*[3] embryonic TC lumen phenotypes range in expressivity from complete absence of subcellular lumen to different lengths of shorter lumina (*Figure 1M,N*). When quantified in detail, out of the 62.5% TCs that showed a luminal phenotype, 36% of TCs did not elongate a subcellular lumen at all (types III and IV) and 64% failed to accomplish a full-length lumen (types I and II) (n = 25) (*Figure 1M,N*).

Recently, it has been reported that endocytosis is involved in subcellular lumen formation (*Mathew et al., 2020*). Bazooka (Baz), the *Drosophila* Par3, which is mainly associated with the apical membrane, has been shown to accumulate at the tip of the TC during lumen formation (*Gervais and Casanova, 2010*; *Mathew et al., 2020*). When we analysed this Baz accumulation in *shot*[3] mutants, we could detect that it was very disrupted and no longer localised at the TC tip (*Figure 1—figure supplement 1*).

Taken together, these results indicated that Shot is involved in de novo subcellular lumen formation and elongation.

### Shot overexpression induces extra-subcellular branching independently of the centrosome

Having observed that Shot was necessary for subcellular lumen formation and extension, we hypothesised that Shot overexpression (ShotOE) would induce extra-subcellular branching events. Indeed, analysis of long-isoform ShotOE (*shotA-GFP*) in tracheal cells revealed that increasing Shot concentrations induced extra-subcellular lumina (ESL) in GB and DB TCs (*Figure 2A–C,J*). Since MTs and actin are essential for subcellular lumen formation (*Gervais and Casanova, 2010*), we then asked whether supernumerary luminal branching was due to the MT- or the actin-binding domains present in the Shot molecule (*Bottenberg et al., 2009*; *Voelzmann et al., 2017*). To this end, we overexpressed an isoform of Shot (ShotC-GFP) with a deletion of the first calponin domain (*Figure 2K,L*), resulting in a shorter actin-binding domain (ABD), which binds actin very weakly or not at all (*Lee and Kolodziej, 2002a*; *Lee and Kolodziej, 2002a*). The tracheal overexpression of *shotA* induced phenotypes in 95% of the embryos (n = 20), with an average of two TC bifurcations *per* embryo (n = 400). *shotC* overexpression induced phenotypes in 90% of the embryos (n = 20), with an average of two TC bifurcations *per* embryo (n = 400) (*Figure 2D–F and G*). In both cases approximately 15% of all TCs analysed displayed an ESL phenotype (*Figure 2J*). In all cases, we could detect more MT-bundles in TCs, associated with the ESLs (*Figure 2A–F* and *Figure 2—figure supplement*

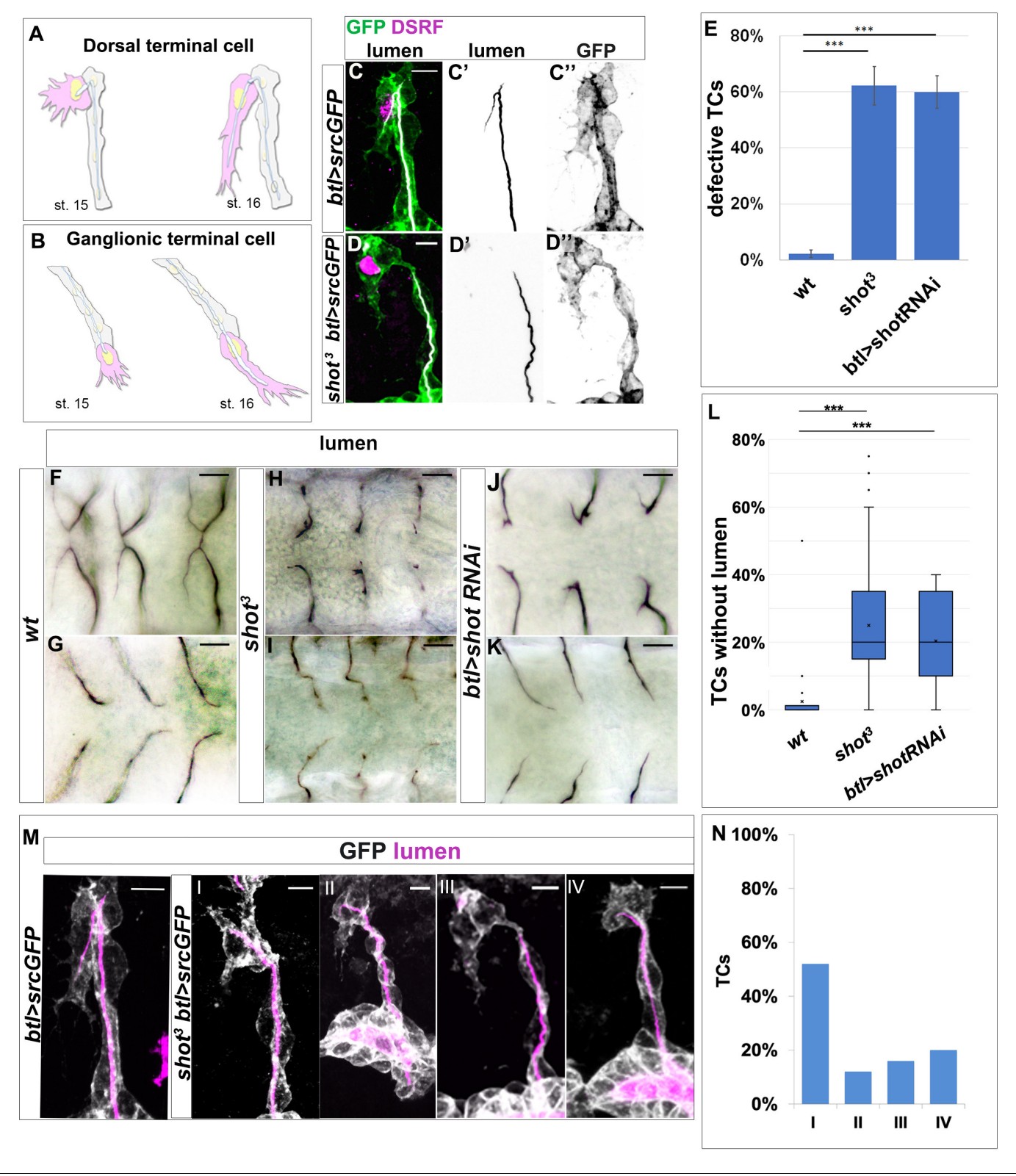

**Figure 1.** shot loss-of-function induces defects in subcellular lumen formation. (A–B) Representation of dorsal and ganglionic TCs from embryonic st.15 to st.16 (DB and GB in grey, TC in pink). At st.15, the TC (cytoplasm in pink, nucleus in yellow, basal membrane in grey, apical membrane in blue and lumen in white) emits filopodia in the direction of cell elongation; apical membrane grows in the same direction giving rise to the outline of the subcellular lumen. At the same time the subcellular lumen is filled of chitin (white). At the end of st.16 the TC is elongated and the subcellular lumen is

*Figure 1 continued on next page*

Figure 1 continued

formed. (C–D) DBs at st.15 of *btl >srcGFP* (control) and *shot³; btl >srcGFP* fixed embryos stained with GFP to visualise tracheal cells, green in C and D, grey in C'' and D'', CBP to visualise the lumen, white in C and D black in C' and D' and DSRF in magenta. Anterior side is on the left and dorsal is up, scale bars 5 µm. (E) Quantification of total defective TCs in *btl >shotRNAi* (60%), *shot³* (62.5%) and *wt* (2.25%) n = 20 embryos, 400TCs. Error bars are ± SEM and asterisks represent a p-value<0001. Statistics by two-tailed Student's *t*-test. (F–K) DBs (F-J dorsal view) and GBs (G-K ventral view) of fixed embryos stained with anti-Gasp antibody at st.16 of *wt* (F and G), *shot³* (H and I) and *btl >shotRNAi* (J and K) (L) Quantification of total TCs (genotype indicated) without subcellular lumen (*wt* 1.34% n = 400, *shot³* 25% n = 400, *btl >shotRNAi* 20%n = 300). *** p-value<0001. Statistics by two-tailed Student's *t*-test. Scale bars 10 µm. (M–N) Different types of TC mutant phenotypes were produced in absence of Shot as observed in detail by confocal microscopy. (M) Dorsal branches of *btl >srcGFP* control and *shot³* embryos stained with GFP (grey) to visualise membrane and CBP (in magenta) to visualise the lumen. Anterior side is on the left and dorsal side is up. Scale bars 5 µm. (I) TC partially elongated with formed lumen but with wrong directionality (52%); (II) the elongation was stopped prematurely and a primordium of subcellular lumen was formed (12%); (III) the cell elongated partially but the lumen was completely absent (16%); and (IV) the cell was not able to elongate and the lumen was completely absent (20%). Types III and IV were quantified in L as TCs without lumen. (E) Detailed quantification, by confocal microscopy, of the different types of TC mutant phenotypes reported as I-IV (n = 25 TCs).

The online version of this article includes the following source data and figure supplement(s) for figure 1:

**Source data 1.** Quantification of shot loss-of-function defects in subcellular lumen formation.
**Figure supplement 1.** Par3 vesicles are mislocalised in *shot* mutant TCs.

1). ShotA-GFP and ShotC-GFP displayed different localisations within the TC. Full-length ShotA-GFP localisation can be detected at the cell-junctions, around the crescent lumen, in MT-bundles, and throughout the cytoplasm, whereas ShotC-GFP localised more to the MT/lumen region, in agreement with the lack of actin-binding capability of ShotC isoform (*Figure 2—figure supplement 1*). Interestingly, we observed a highly ramified subcellular lumen when higher amounts of ShotC were expressed in tracheal cells (*Figure 2G*) suggesting that the effect of ShotOE in subcellular lumen branching was dosage dependent.

Tracheal overexpression of *shotC* phenocopied that of *shotA* in inducing ESLs (14.63% and 15.5% ESL respectively, *Figure 2D–F and J*), suggesting that the ABD is not necessary for the induction of additional luminal branching events. In order to clarify this, we used two other isoforms of Shot: *shotΔCtail*, lacking the C-terminal MT-binding domain, and *shotCtail,* a truncated form containing only the C-terminal MT-binding domain (*Alves-Silva et al., 2012*; *Figure 2K,L*). Whereas overexpressing *shotΔ-Ctail* in TCs we could only detect a branching phenotype in 1,5% of TCs analysed (n = 400), (*Figure 2K*), overexpression the C-tail domain alone induced TCs with extra branching in 9.5% of TCs (n = 400) (*Figure 2I*), indicating that the C-tail alone was sufficient to induce ESLs in TCs. Taken together these results using different Shot isoforms, lead us to conclude that the Shot MT-binding domain alone is sufficient for the extra branching events observed in ShotOE TCs.

ESLs were previously observed when higher numbers of centrosomes were present in TCs (*Ricolo et al., 2016*). We therefore asked if the observed extra branching phenotypes could be due to supernumerary centrosomes induced by ShotOE in TCs. Consequently, we quantified the number of centrosomes in the TCs of ShotOE embryos. In *wt* TCs we detected an average of 2.3 ± 0.5 (n = 33) centrosomes per TC, and in ShotOE 2.2 ± 0.2 centrosomes per TC (n = 33) (*Figure 3A–C*). In both conditions, and as previously described (*Ricolo et al., 2016*), this centrosome pair was detected at the apical side of the TCs (*Figure 3A,B*). Besides, analysing ShotOE TCs at embryonic st.15, (n = 16) we could detect that the ESL arose from the pre-existing subcellular lumen, distally from the centrosome pair (*Figure 3B'* arrow). These data indicate that ShotOE did not change TC centrosome number and induced ESL by a distinct mechanism from centrosome duplication.

*Regulator of cyclin A1 (Rca1)* is the *Drosophila* ortholog of vertebrate Emi1 and a regulator of APC/C activity at various stages of the cell cycle (*Grosskortenhaus and Sprenger, 2002*). *Rca1* mutants have supernumerary centrosomes at the TCs and develop ESLs at embryonic stages (*Ricolo et al., 2016*). In contrast with ShotOE alone (*Figure 3B'*), in *Rca1* mutants the bifurcated subcellular lumen arose from the apical junction and continued to extend during TC development (*Ricolo et al., 2016*). When we analysed the luminal origins in *Rca1*, ShotOE conditions, both types of ESL where detected in the same TC in 25% of the cases (n = 12). In the same TCs two types of ESL were generated, one from the apical junction and another sprouted from the pre-existing lumen distally from the junction (*Figure 3D*, asterisks). In addition, the effect of *Rca1* LOF and ShotOE was additive in producing TCs with a multiple-branched subcellular lumen (*Figure 3I*). These

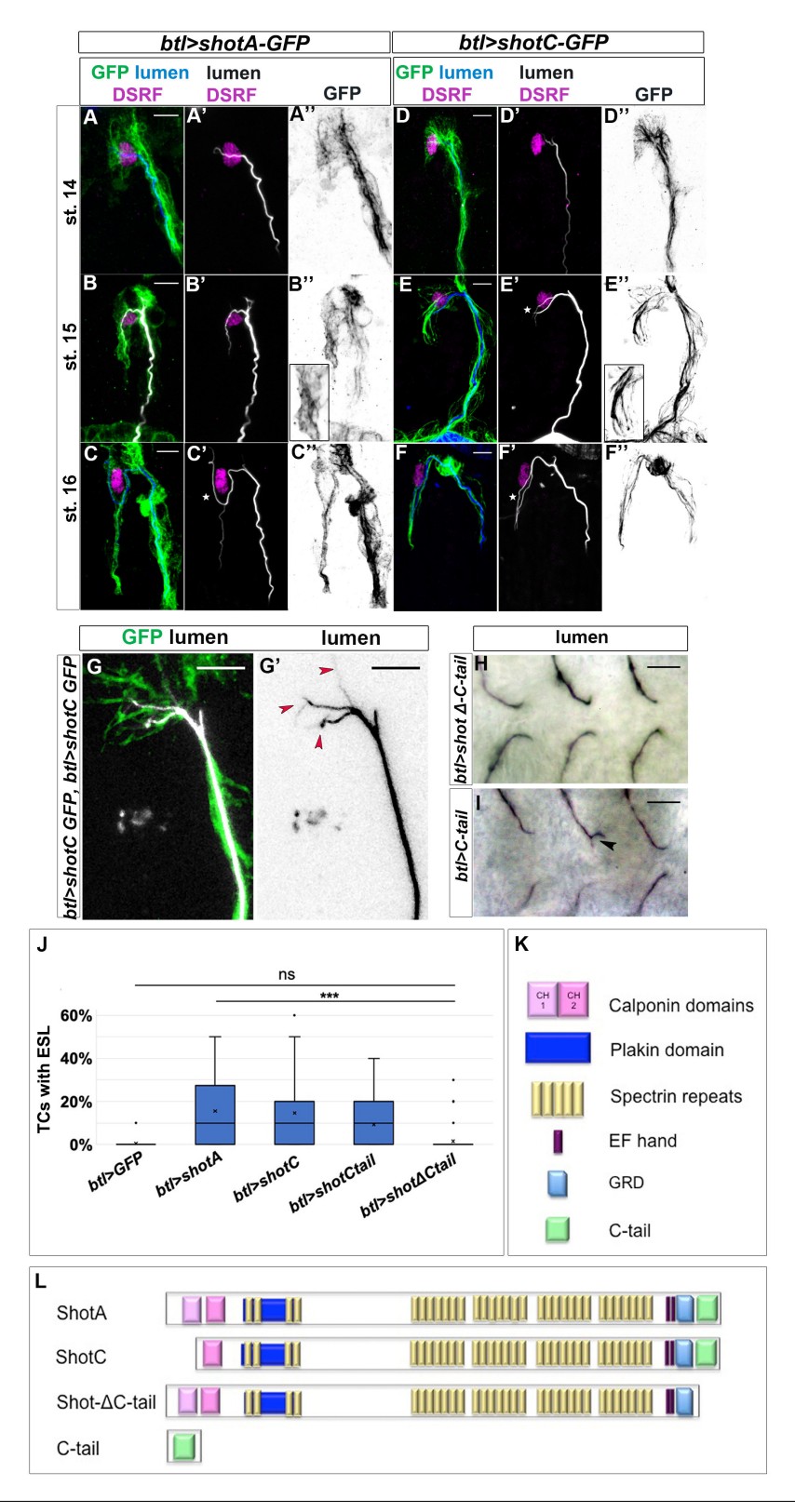

**Figure 2.** *ShotOE* induces luminal branching through its microtubule-binding domain. Lateral view of DB tip cells from st.14 to st.16, of *btl >shotA*-GFP embryos (A–C) and *btl >shotCGFP* (D–F). Embryos were stained with GFP (green in A-F and grey A''-F'') to visualise Shot-GFP, DSRF to mark the TC nuclei (in magenta) and CBP to stain the chitinous lumen (blue in A-F and white A'-F'). Both overexpressing conditions induced ESLs (white stars). Note the GFP was more diffuse in the cytoplasm of the TCs of embryos overexpressing *shotA*, and more organised in bundles in the TCs overexpressing

*Figure 2 continued on next page*

*Figure 2 continued*

*shotC.* Anterior side of embryo is on the left and dorsal side up. Scale bars 5 µm. (**G**) ESL induction by ShotOE is dosage sensitive. Example of dorsal TC of an embryo overexpressing two copies of *btl >shotC-GFP*, stained with anti-GFP (green) and CBP (white in G, black in G'). Red arrows indicate extra-subcellular lumen branching. Note that the extra-subcellular lumina are very thin and they follow Shot positive bundles detected with GFP. Anterior side is on the left, dorsal midline is on the top. Scale bars 5 µm. (**H, I**) Tips of GB TCs from *btl >shotΔCtail* embryos with a single subcellular lumen each (**H**) and *btl >C-*tail (**I**) in which one TC is bifurcated; stained with anti-Gasp (ventral view, anterior side of the embryo is on the left). Scale bars 10 µm. (**J**) The C-tail domain is involved in ESL formation. Percentage of TCs displaying ESLs in embryos overexpressing *GFP*, *shotA*, *shotC*, *C-tail*, and *shotΔCtail* in the tracheal system (n = 400 TCs all genotypes except *btl >shotC* where n = 800). *** p-value<0.001; **ns** refers to a p-value>0.1. Statistics by two-tailed Student's *t*-test. There was no significant difference between overexpression of *shotΔCtail* and GFP alone. (**K**) Schematic representation of spectraplakin protein domains and (**L**) the different Shot constructs used in this study.

The online version of this article includes the following source data and figure supplement(s) for figure 2:

**Source data 1.** Quantification of ShotOE phenotypes in subcellular lumen formation.
**Figure supplement 1.** ShotOE does not perturb the TC cytoskeletal organisation.

morphological ESL differences suggested that *Rca1* and *shot* operate in different ways in the de novo formation and branching of the subcellular lumen.

## Shot associates with stable microtubules and actin

Spectraplakin expression is critical in cells that require extensive and dynamic cytoskeleton reorganisation, such as epithelial, neural, and migrating cells. Loss of spectraplakin function leads to a variety of cellular defects due to disorganised cytoskeletal networks (*Hahn et al., 2016*). In a plethora of tissues and in cultured S2 cells, Shot can physically interact with different cytoskeletal components (*Applewhite et al., 2010*; *Lee and Kolodziej, 2002b*; *Sanchez-Soriano et al., 2009*). Therefore, we investigated Shot localisation and its interaction with MTs and actin in control TCs.

We analysed live embryos using time-lapse imaging and observed that Shot localisation was extremely dynamic throughout subcellular lumen formation. We could detect Shot in the apical TC junction as well as extending together with the growing subcellular lumen (*Video 1* and *Figure 4—figure supplement 1A*). It was apparent that Shot localised dynamically with the growing luminal structures, showing a strong localisation at the middle/tip of the extending TC (*Video 1* and *Figure 4—figure supplement 1A*).

In control conditions actin concentrated strongly at the tip of the TC, but was also detected in the TC cytoplasm, and these different actin populations have been shown to be important for subcellular lumen formation and extension (*Gervais and Casanova, 2010*; *Oshima et al., 2006*). During TC elongation, MTs polymerise from the centrosome pair at the apical junction toward the tip of the cell, reaching the area of high actin accumulation at the migrating tip of the TC (*Gervais and Casanova, 2010*; *Ricolo et al., 2016*). So, we next analysed ShotA and ShotC localisation in relation to the dynamically localised actin core present in the cytoplasm and at the tip of the migrating TC in live embryos (*Videos 2*, *3*, and *4* and *Figure 4—figure supplement 1*). In both GB and DB TCs we could detect a dynamic interaction between the long Shot isoform (ShotA) and actin as detected by Moe::RFP (the ABD of moesin fused to RFP, thereby labelling the actin cytoskeleton of these cells, *Video 4*) or Life-actRFP (*Video 3* and *Figure 4—figure supplement 1C*). ShotA dynamically interacted with different actin populations, namely the actin core and basal, filopodial actin (*Videos 3* and *4* and *Figure 4—figure supplement 1C*). However, as the lumen extended, ShotC fibres extended with the cell, surrounding the growing luminal area, like Shot A, but no strong interaction was detected with the dynamic actin core and with the basal, filopodial actin (*Video 2* and *Figure 4—figure supplement 1B*).

We followed these analyses, observing endogenous Shot in fixed and antibody stained embryos. At early stages, when TCs started to elongate, we detected Shot co-localizing with actin at the tip of the TC (*Figure 4A*). The overlap between Shot and actin was maintained until late st.15 (*Figure 4B*). Then, we examined Shot localisation in relation to MTs. Shot was strongly detected in the TC from early stages of lumen extension and until the end of TC elongation (*Figure 4D–E*). At the beginning of de novo lumen formation, when MTs emanated from the junction/centrosome pair, Shot co-localised with the first sprouting stable MTs (*Figure 4C–D*). The overlap between Shot and stable MTs was strongly observed also at embryonic st.15 when a MT track preceded subcellular lumen

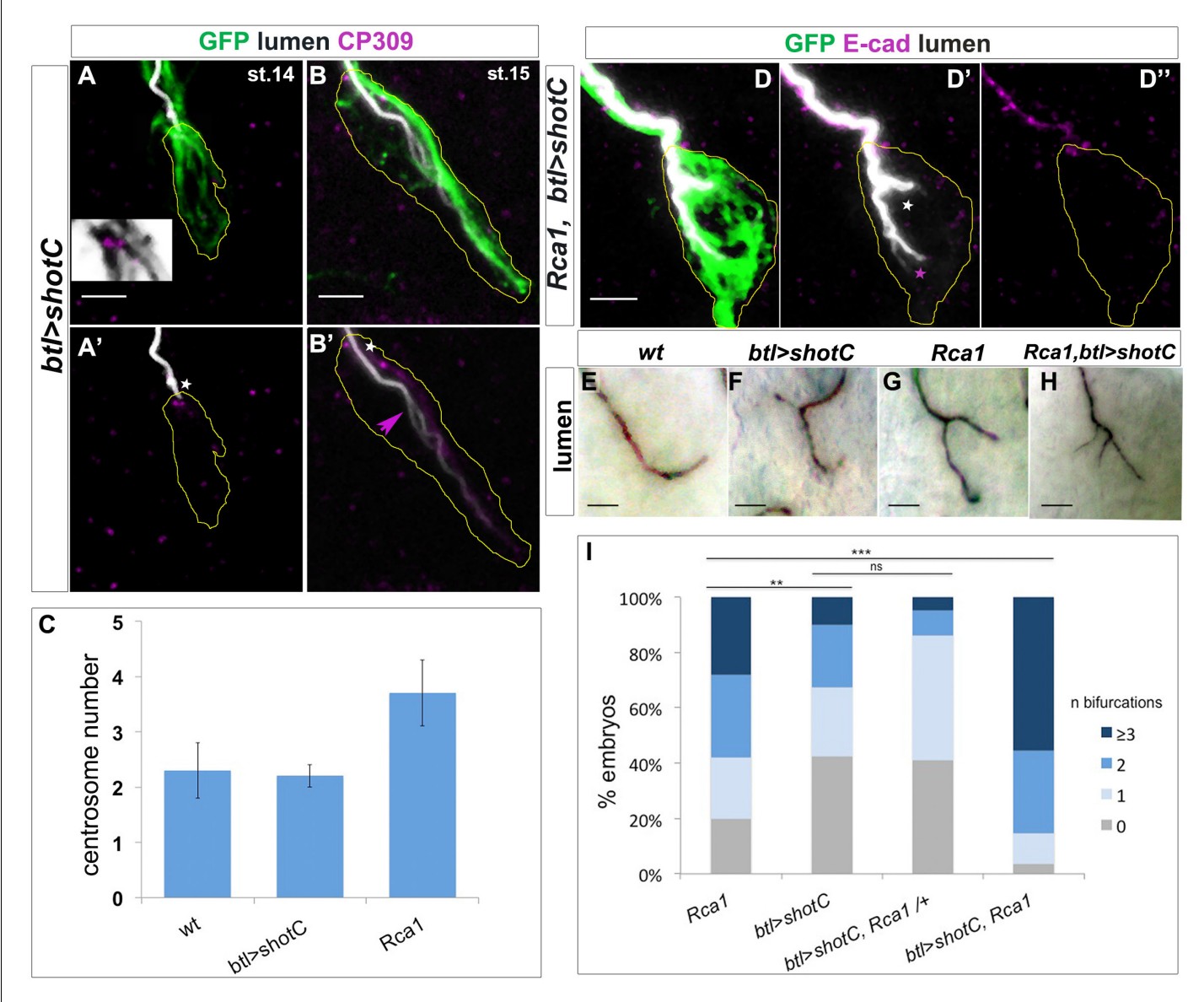

**Figure 3.** ESL induction by *ShotOE* is not associated with centrosome amplification. (A,B) GB TC of a st.14 embryo (A), prior to lumen extension, and st.15 when a bifurcated lumen can be detected in ShotOE conditions. (A,B) *btl >shotC*-GFP embryos stained with CBP to mark lumen (white), anti-GFP to visualise Shot (green) and anti-CP309 antibody to mark centrosomes (magenta); the outline of TCs is drawn in yellow. The box in A is a magnification of the apical side, showing the TC centrosome pair (in magenta) and GFP positive Shot bundles (in grey) emanating from centrosomes. White stars indicate centrosomes apically localised. Note in B' the subcellular lumen (magenta arrow) bifurcated in a point downstream from the centrosome pair. Anti-CP309 antibody stains all centrosomes throughout the embryo and not just in TCs and cross-reacts with an unidentified antigen at the subcellular lumen. (C) Quantification of centrosome number in *wt, btl >shotC* and *Rca1* embryos ± SEM. (D) GB tips from *Rca1; btl >shotC*-GFP embryos at st.15, stained with CBP (in white) to visualise the lumen and E-cadherin (magenta) to recognise the TC apical junction. Anterior side of the embryo is on the left and ventral is down. Scale bar 2 μm. In these cases, it is possible to detect two types of luminal bifurcations: one from the apical junction (white arrow), caused by *Rca1* mutant supernumerary centrosomes, and another one arising from a pre-existing lumen, induced by ShotOE. (E–H) Details of GB TCs at st.16 from embryos stained with anti-Gasp antibody to mark the lumen. (E) *wt* TCs with a single lumen each; (F) *btl >shotC* showing subcellular lumen bifurcations; (G) *Rca1* showing subcellular lumen bifurcations; (H) *Rca1; btl >shotC* showing a multi-branched subcellular lumen. Anterior side of the embryo is on the left ventral midline is down. Scale bars 5 μm. (I) Quantification of the number of bifurcations (GB TCs) per embryo of the indicated genotype. 'n bifurcations' is the number of ESL per TC.

The online version of this article includes the following source data for figure 3:

**Source data 1.** Quantification of centrosome number and luminal phenotypes.

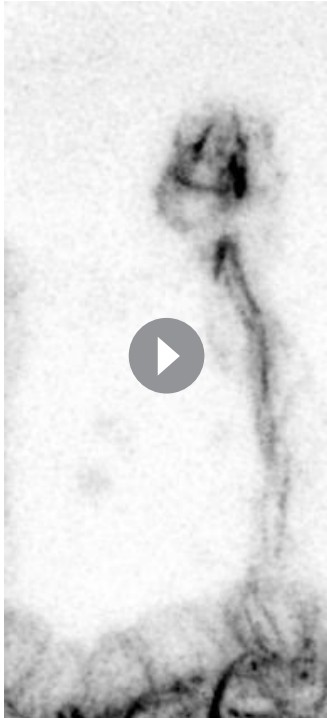

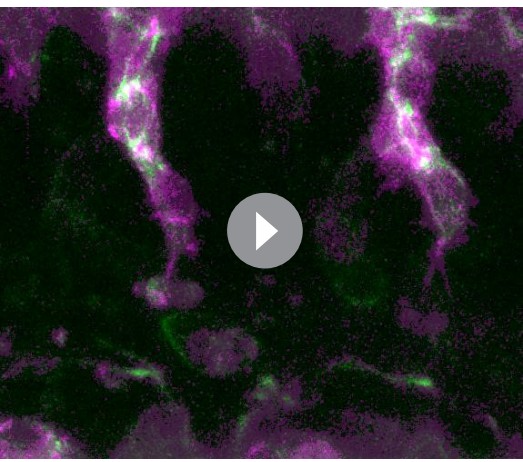

**Video 2.** Shot localises with the extending subcellular lumen in TCs during subcellular lumen extension. In vivo Shot colocalisation with actin during lumen formation in a wild-type dorsal branch (DB) TC. Time-lapse images of a *wt* embryo expressing btlGAL4UASShotC-GFP; btl::moeRFP visualised from a dorsal view. Note the colocalisation of Shot with actin the apical junction of the TC. As the cell extends, ShotC localises to the growing lumen and there is no detectable strong association with the actin core and filopodial actin. Frames were taken every 2 min for 3.3 hr.
https://elifesciences.org/articles/61111#video2

**Video 1.** Shot localisation within the TCs is dynamic and accompanies the growing subcellular lumen. In vivo Shot localisation during lumen formation in two wild-type ganglionic branch (GB) TCs. Time-lapse images of a *wt* embryo expressing btlGAL4UASShotC-GFP visualised from a dorsal view. Note the accumulation of Shot at the apical junction of the TC and subsequently in association with subcellular lumen extension. Frames were taken every minute for 3.5 hr.
https://elifesciences.org/articles/61111#video1

detection (*Figure 4D*). At st.16, both Shot and stable MTs localised to the apical side of the TCs in the area surrounding the subcellular lumen (*Figure 4E*).

Shot localisation within the TC suggested that the spectraplakin localised with stable MTs all around the nascent lumen and with the actin at the tip of the TC, during the time of cell elongation and subcellular lumen formation. This suggests that Shot mediates the crosstalk between these two cytoskeletal components, helping their stabilisation and organisation during subcellular lumen formation and growth (*Figure 4F*).

## Absence of shot leads to disorganised microtubules and actin

We then asked how actin and MTs were localised and organised in *shot*[3] mutant embryos. We analysed the different types of TC mutant phenotypes ranging from cases in which the TC did not elongate and the subcellular lumen was not formed, to cases in which the TC was able to elongate and form the lumen albeit not to the levels in control embryos (*Figure 5*). In all cases, we found defects in both MTs and actin accumulation in mutant TCs.

Considering actin localisation, in control embryos at early st.16, Moe::RFP detecting actin was strongly localised at the tip of the TC, in front of the tip of the growing lumen (86% of TCs analysed, n = 21). Moreover, a few spots of actin were detectable in the cytoplasm, around the subcellular lumen (*Figure 5A* and *Video 2*). In *shot*[3], we observed reduced actin accumulation at the TC-tip and an increase of scattered spots into the cytoplasm (86% of TCs analysed, n = 23) (*Figure 5B–D*), indicating that Shot contributed to TC actin organisation.

Regarding MT-bundles, we observed stable MTs organised in longitudinal bundles around the subcellular lumen in control TCs (*Figure 5E*). In *shot*[3] TCs (n = 20), we detected MT-bundle defects. In particular, we observed that when the TC was not elongated, MT-bundles no longer localised to the apical region and seemed to be fewer than in *wt* (*Figure 5F*). A general disorganisation in MT-

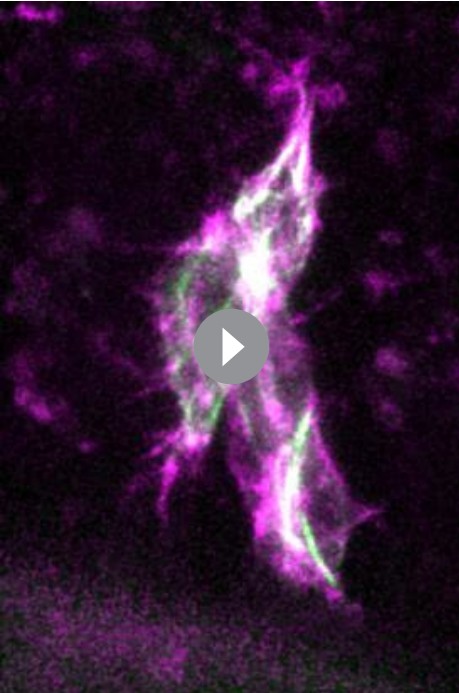

**Video 3.** Shot localises with actin in TCs during the early steps of subcellular lumen extension. In vivo Shot long-isoform colocalisation with actin during lumen formation in a wild-type dorsal branch (DB) TC. Time-lapse images of a *wt* embryo expressing btlGAL4UASShot-A-GFPUASlifeActRFP visualised from a dorsal view. Note the colocalisation of Shot with actin at the apical junction of the TC and subsequent association with the actin core and filopodial actin during subcellular lumen extension. Frames were taken every 40 s for 32 min.
https://elifesciences.org/articles/61111#video3

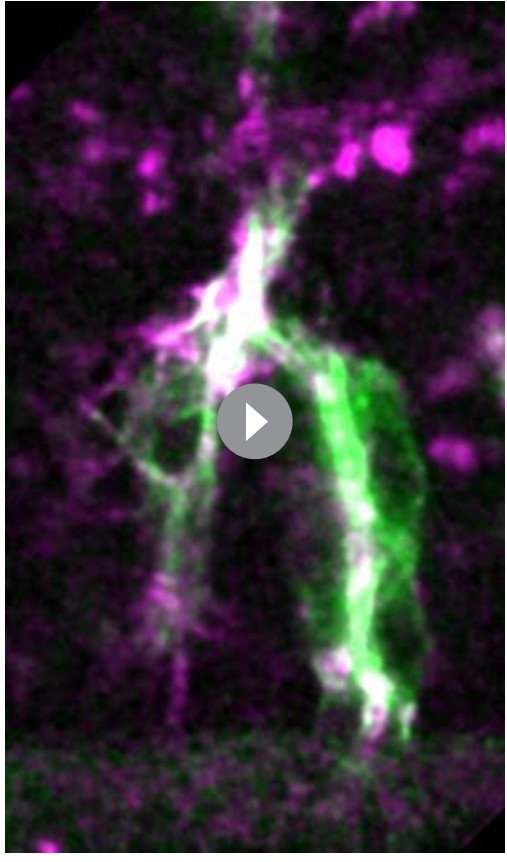

**Video 4.** Shot localises with actin in TCs during later steps of subcellular lumen extension. In vivo Shot long-isoform colocalisation with actin during lumen formation in a wild-type dorsal branch (DB) TC. Time-lapse images of a *wt* embryo expressing btlGAL4UASShotA-GFP; btl::moeRFP visualised from a dorsal view. Note the colocalisation of Shot with filopodial actin during subcellular lumen extension. Frames were taken every 30 s for 36 min.
https://elifesciences.org/articles/61111#video4

bundles in respect to the control was also observed in TCs partially able to elongate a subcellular lumen (*Figure 5G,H*).

These analyses, taken together with the previous analysis of Shot localisation in control TCs, suggested a spectraplakin role in organizing/stabilizing both MTs and Actin accumulation in the TC.

## Subcellular branching depends on both actin and microtubule-binding domains of shot

In order to analyse how the different domains of Shot affected luminal development and branching, we expressed different isoforms of Shot in *shot³* mutant TCs. As described previously, *shot³* embryos displayed a variable expressivity in TC phenotypes. To simplify the quantification of the rescue experiments, we took in consideration the most severe luminal phenotype: the complete absence of a subcellular lumen. In *shot³*, we quantified that 22% of TCs (at the tip of GBs and DBs) did not develop a subcellular lumen at all (*Figure 1H,I* and *Figure 6B,J*). Targeted expression of full-length ShotA in the trachea of *shot³* mutant embryos was able to rescue the subcellular lumen phenotype to the level of only 6% of the TCs analysed (n = 200) not developing a subcellular lumen (*Figure 6C*).

We then proceeded to molecular dissect the function of Shot in TCs. To do so, we used the three different constructs Shot: ShotC, ShotΔCtail and ShotCtail (*Figure 2L*). When we expressed ShotC in the tracheal TCs we found that 20% of TCs analysed (n = 200), had TCs with no lumen (*Figure 6D*

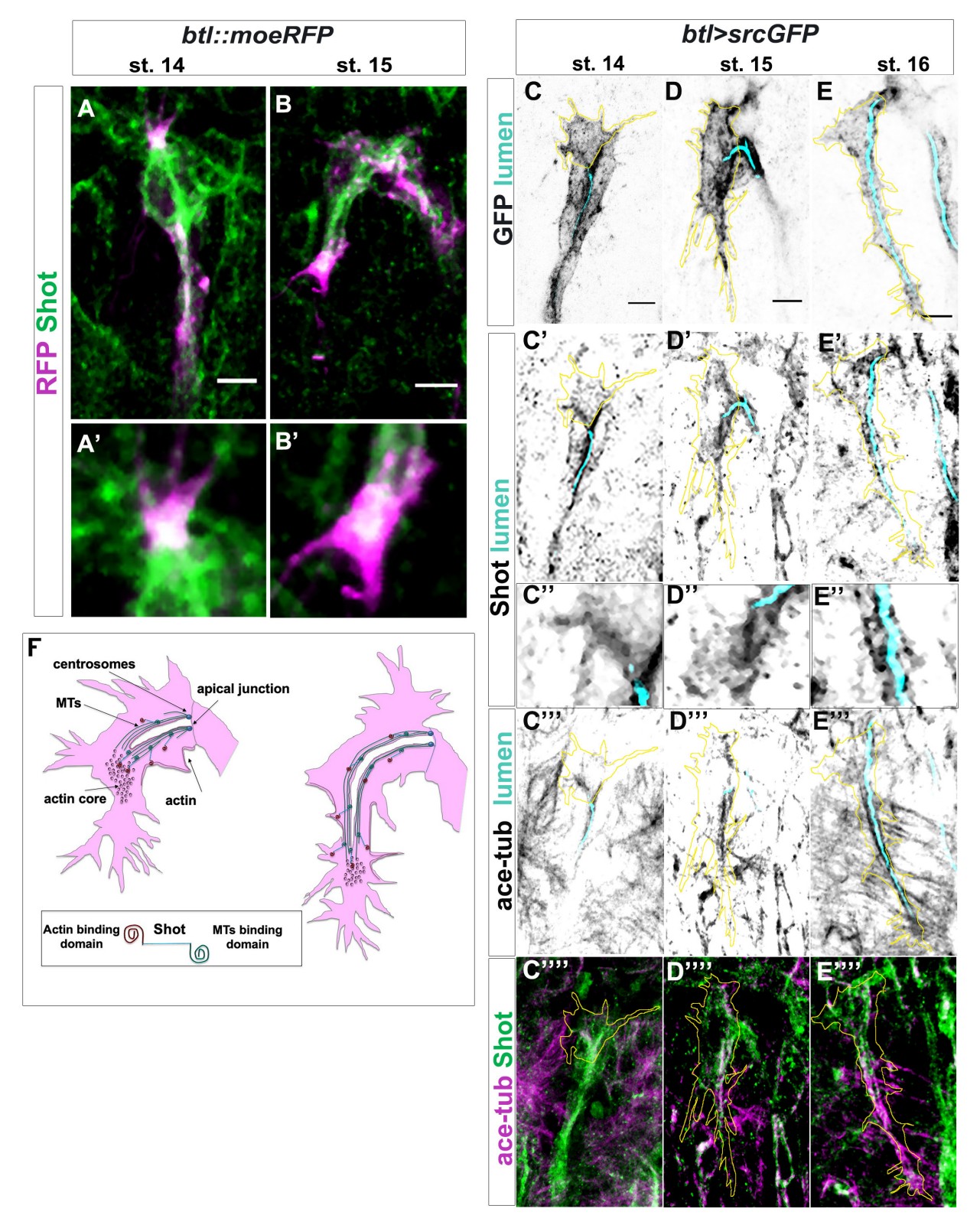

**Figure 4.** Shot colocalises with TC cytoskeletal components. (A–B) Endogenous Shot, detected by an antibody in btl::moeRFP embryos, colocalised with actin during TC development. Tip of dorsal branches from st. 14 to late st. 15 of *btl::moeRFP* embryos stained with RFP (magenta) and Shot (green). In the magnification of the tip of the TCs (A'–B') note Shot and RFP co-localisation in both embryonic stages. (C–E) Endogenous Shot accumulated around stable microtubules during subcellular lumen formation. The whole TC morphology changes overtime. This is a developing

*Figure 4 continued on next page*

*Figure 4 continued*

structure and both cell-shape and the cytoskeleton are changing throughout its development. Here we provide snapshots of different stages showing the beginning, middle and final stages of TC lumen formation. Dorsal TCs from fixed embryos *btl >srcGFP* stained with Shot and acetylated-tubulin antibodies and fluostain to detect chitin, from st.14 to st.16. GFP staining is showed in grey and cell contour in yellow (C–E), endogenous Shot is shown in grey in panels C'- E '' (C''- E'' are magnification of C'- E') and green C''''- E''''. Acetylated tubulin is in grey in C'''- E''' and magenta in C'''' -E''''. The chitinous lumen was detected with fluostain, represented in cyan (C– E''''). Acetylated tubulin and Shot are both accumulated ahead of the subcellular lumen at earliest stages (st. 14–15) and around the subcellular lumen at later stages (st.16). Note that co-localisation between acetylated tubulin and Shot is detected in the TCs. Anterior side is on the left, dorsal midline is up. Scale bar 5 μm. (F) Schematic representation of dorsal TC development from st.15 to st.16. Basal membrane in grey, apical membrane in light blue, subcellular lumen in white, the actin network in red and MTs are in green. Between st.14 and st.15 actin dots mature in an actin core in front of the tip of the subcellular lumen in formation that is surrounded by microtubules. Shot (represented on the bottom of the figure) was detected both inside the actin core and surrounding the lumen where stable MTs are organised. The online version of this article includes the following figure supplement(s) for figure 4:

**Figure supplement 1.** Shot is dynamically localised in TCs and interacts with actin.

*and J*), suggesting that the ABD domain is necessary for the correct de novo luminal morphogenesis.

We next expressed *shotC-tail* in order to address whether the Shot MT-binding domain alone could restore subcellular lumen formation. We observed that 24% of TCs analysed at the tip of GBs and DBs (n = 250) were still not able to form a subcellular lumen (*Figure 6E and J*), suggesting that the tracheal expression of *shotC-tail* was not enough to rescue the null phenotype. Finally, we expressed *shot-ΔC-tail* to test whether Shot without the MT-binding domain could restore subcellular lumen formation. We observed that 16% of TCs analysed at the tip of GBs and DBs (n = 250) were still unable to form a subcellular lumen (*Figure 6F and J*). Taken together, these analyses suggested that full-length isoform A, allowing Actin-MT crosslinking is necessary for correct de novo subcellular lumen formation.

In order to further test the hypothesis that full-length Shot is needed to correctly form a subcellular lumen, we analysed *shot*[kakP2] mutant phenotype. This allele carries an insertion of a transposable element into the intron between the second and the third transcriptional start site of *shot* abolishing all isoforms containing the first Calponin domain (CH1) and interfering with Shot actin-binding activity (*Bottenberg et al., 2009*). The penetrance and expressivity of the phenotype observed in *shot*[kakP2] TCs was very similar to *shot*[3] null allele with 18% of these (n = 600; 300 ganglionic and 300 dorsal TCs) not forming a subcellular lumen at all (*Figure 6G,J*). In addition, *shot*[kakP2] TCs display the same MT and actin disorganisation phenotypes as *shot*[3] TCs (*Figure 6—figure supplement 1B–D*). Phenotypic data from *shot*[kakP2] together with data from transgenic rescues with the *ShotC* construct, lacking the CH1 domain, indicate that Shot full length is required for de novo subcellular lumen formation.

Since the actin and MT-binding domains were shown to be necessary for the proper formation of a subcellular lumen, we asked whether it was necessary to have both domains in the same protein or if simply the independent presence of these domains was enough to generate a subcellular lumen. To do so, we generated transheterozygous flies expressing two different Shot isoforms, Shot[kakP2] and Shot[ΔEGC]. Shot[ΔEGC] is a truncated protein, lacking the EF-hand, the Gas2 and the C-tail domains of Shot, leading to complete loss of the MT-binding activity (*Takács et al., 2017*). The analysis of *shot*[ΔEGC] mutant TC phenotypes revealed that 18% of TCs (n = 400; 200 ganglionic and 200 dorsal TCs) did not develop a TC lumen at all (*Figure 6H,L*) and that *shot*[ΔEGC] mutant TCs display the same MT and actin disorganisation phenotypes as *shot*[3] TCs (*Figure 6—figure supplement 1E–G*).

In *shot*[ΔEGC]/*shot*[kakP2] transheterozygous embryos, Shot molecules contained exclusively either the CH1 or the C-tail, but neither molecule had actin- and MT-binding activity simultaneously. These embryos displayed the same phenotype as either homozygous mutant (18% TCs with no lumen, n = 400) (*Figure 6I*), indicating that both the actin- and the MT-binding domains need to be present in the same Shot molecule for proper TC subcellular lumen formation.

Taken together these results indicate that Shot is able to mediate the crosstalk between MTs and actin during subcellular lumen formation, via its MT and actin-binding domains and that these have to be present in the same molecule for proper subcellular lumen formation.

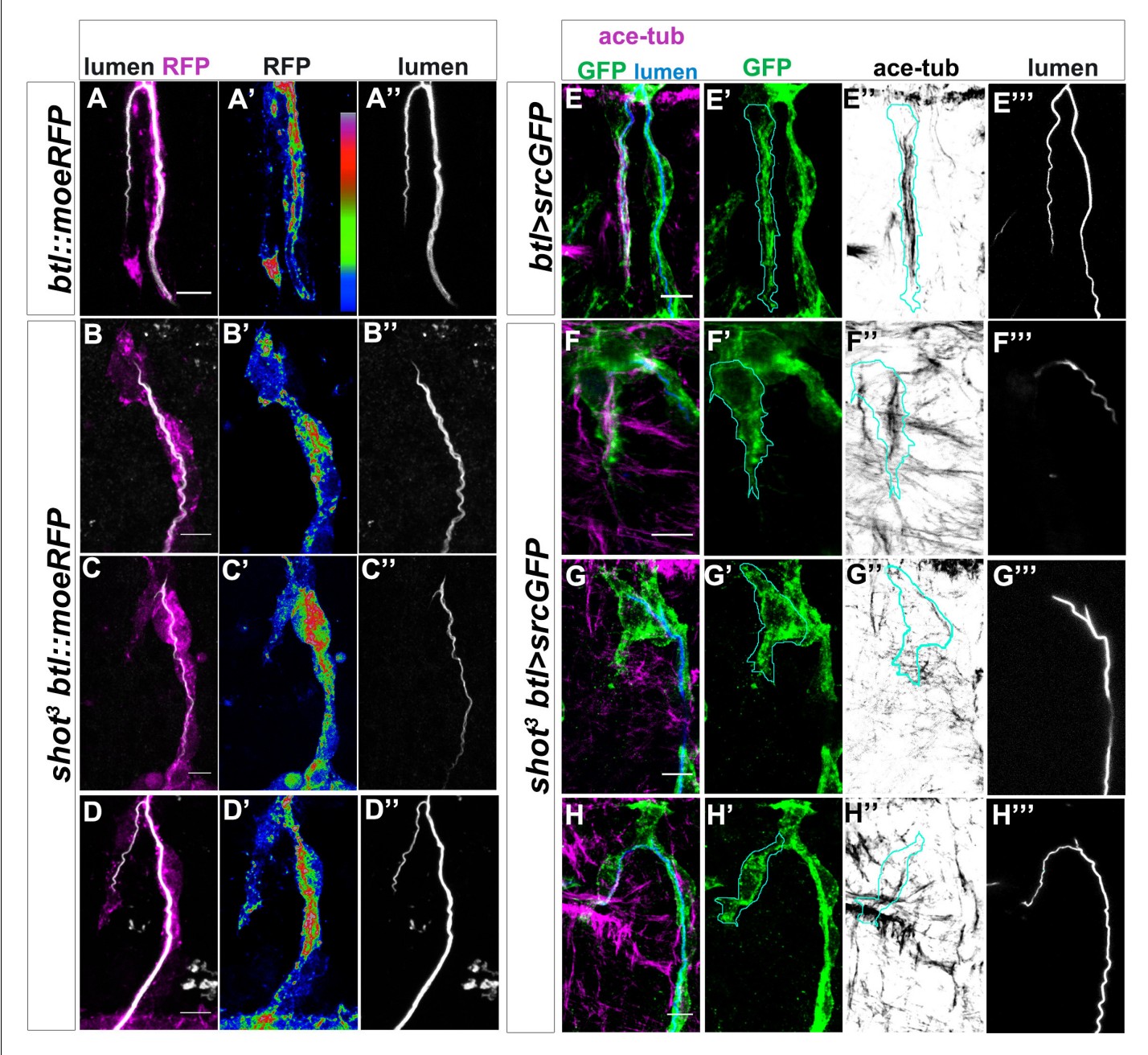

**Figure 5.** Shot LOF lead to disorganised MT-bundles and actin. (A–D) Asymmetric actin accumulation in extended TCs was affected in *shot³* mutant embryos. Dorsal TC from fixed *shot³/+; btl::moeRFP* controls (A) and *shot³; btl::moeRFP* mutant embryos (B–D), stained with RFP (Magenta in A-D or in a colour scale in which blue is low, green is middle and red high intensity in A'- D') and CBP (in white). In *shot* mutant TCs actin appeared affected in its accumulation (B–D). (A) control; (B) when the cell was not elongated and the lumen was not formed; (C) when the cell was partially elongated and the lumen was not formed; (D) when the cell elongated and the lumen was partially formed (D). Note that actin was affected even when the cell was elongated and a lumen was partially formed. (E–H) TC MT-bundles in *shot³*. Dorsal TC from embryo at st.16 control (A) and *shot³* mutant embryos (B–D) stained with GFP (green) acetylated tubulin (in magenta in E-H and in grey in E''-H'') and CBP (in blue in E-H and grey in E'''-H'''). The border of TC was drawn in cyan (E–H''). In all cases, the organisation and the amount of stable MTs was strongly affected, in (F) MT-bundles were observed to be disorganised along the cytoplasmatic protrusion without subcellular lumen and in G and H only a thin track of MTs surrounds the subcellular lumen. Anterior side is on the left and dorsal midline is up. Scale bars 5 μm.

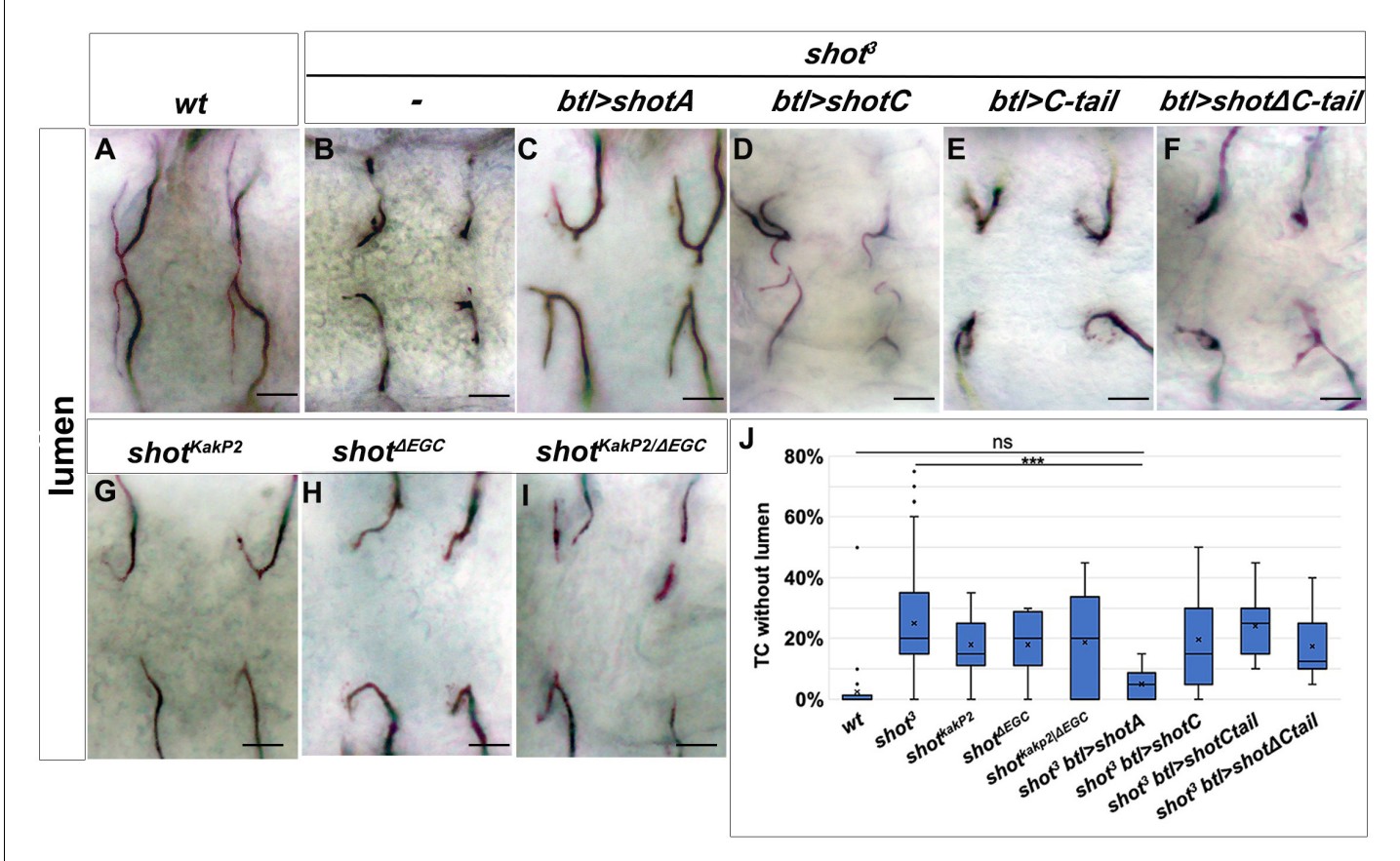

**Figure 6.** Shot Actin- and MT-binding domains are necessary for proper subcellular lumen formation. (A–I) Dorsal branches of st.16 embryos, stained with anti-GASP to visualise the lumen. Genotype is indicated above each panel. Null allele, $shot^3$, rescue experiments (B–F) indicate that both the actin-binding domain (ABD) and the microtubule-binding domain are involved in subcellular lumen formation since the only construct able to rescue the null allele phenotype is the UASShotA (B and J). Both functional domains are needed in the same molecule since mutants affected only in the ABD ($shot^{KakP2}$) or in the microtubule-binding domain ($shot^{\Delta EGD}$) and the transheterozygous $shot^{KakP2/\Delta EGD}$ display the same phenotype as $shot^3$. Scale bars are 10 µm. (J) Quantification of TCs without lumen: wt (n = 820); $shot^3$ (n = 600); $shot^{KakP2}$ (n = 400); $shot^{\Delta EGD}$ (n = 400); $shot^{KakP2/\Delta EGD}$ (n = 320 TCs); $shot^3$, btl >shotA (n = 240); $shot^3$; btl >shotC (n = 240); $shot^3$; btl >shotCtail (n = 240); $shot^3$; btl >shot$\Delta$Ctail (n = 240). *** p-value<0.001; ns refers to a p-value>0.1. Statistics by two-tailed Student's t-test. Only ShotA significantly rescued the $shot^3$ TC luminal phenotype.

The online version of this article includes the following source data and figure supplement(s) for figure 6:

**Source data 1.** Quantification of rescue experiments and different shot mutant phentotypes and transheterozygous combinations.

**Figure supplement 1.** Asymmetric actin accumulation is affected in $shot^{kakp2}$ and $shot^{\Delta EGC}$ Dorsal TCs from btl::moeRFP.

## Increased levels of shot are induced in TCs by DSRF

The TC-specific transcription factor *bs*/DSRF is important for TC specification and growth, and has been suggested to regulate the transcription of genes that modify the cytoskeleton (*Guillemin et al., 1996*; *Olson and Nordheim, 2010*). Considering the luminal phenotypes associated with *bs* LOF in TCs and the role of MTs in subcellular luminal formation, we asked whether *shot* expression in TCs could be regulated by DSRF.

In order to test this, we searched in silico for DSRF binding sites in the promoter regions of all *shot* isoforms using the Matscan software (*Blanco et al., 2006*) and the reported position weight matrix (PWM) corresponding to SRF (*Khan et al., 2018*; *Supplementary file 1*). We found seven regions with at least one putative binding site (binding score larger than 70% of maximum value) within 2000 bases of the *shot* annotated TSS (*Figure 7F* and *Supplementary file 1*). These regions mapped to the locations of known Shot promoters (*Figure 7F*; *Hahn et al., 2016*). We then asked if lower Shot protein levels could be detected in *bs* mutant TCs. Indeed, when analysing *bs* in comparison to *bs/+* TCs, we could detect lower levels of endogenous Shot protein and *shot* mRNA

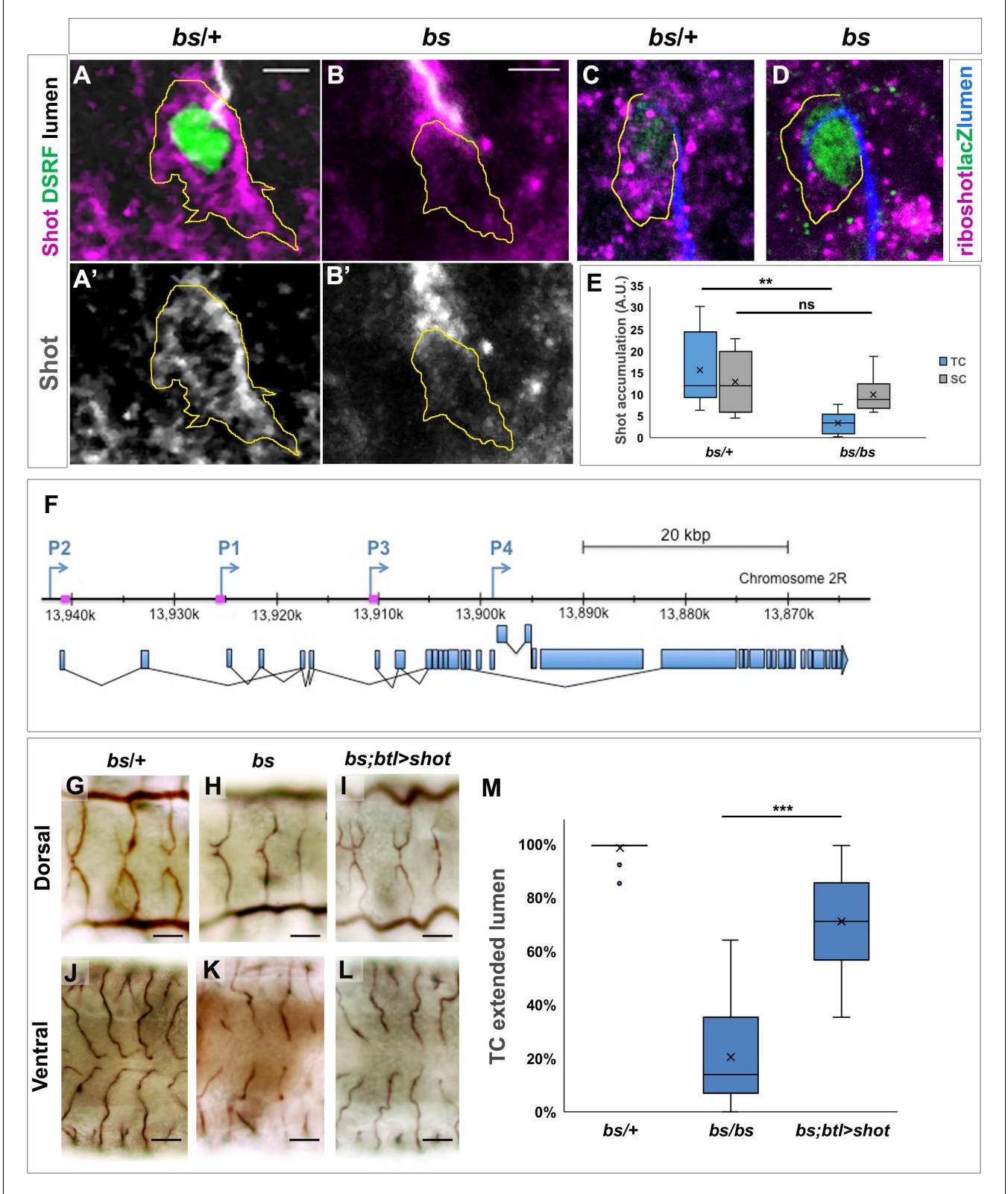

**Figure 7.** Shot expression is regulated by DSRF in TCs GB TC at st.15 from *bs* heterozygous controls. (A) and homozygous (B) mutant embryos, stained with Shot (magenta in A and B, grey in A' and B'), DSRF (green) antibodies and CBP (grey). In yellow, the outline of the TCs. Shot protein was less accumulated in TCs from homozygous *bs* embryos (B, B' and E) n = 9 TCs. Scale bars are 5 µm. (C,D) DB TCs from *bs* heterozygous (C) control and homozygous (D) mutant embryos from wholemount FISH with a ribo-shot probe (magenta), stained with anti-betagal (green, to detect the DSRF

*Figure 7 continued on next page*

*Figure 7 continued*

enhancer trap lacZ expression) and CBP (blue) to mark the lumen. LacZ expression is higher in mutant embryos, homozygous for the lacZ P-element insertion (**D**). The yellow line marks the TC outline. Lower levels of *shot* mRNA were detected in *bs* mutant TCs when compared to control TCs (n = 8). Scale bars are 5 μm. (**E**) Quantification of Shot protein in control and mutant TCs and stalk cells (SC). Quantification of raw integrated pixel density in arbitrary units measured in Fiji in the TC and attached SC in each embryo. \*\*p<0.01; **ns** refers to a p-value>0.1. Statistics by two-tailed Student's *t*-test. (**F**) P1, P2 and P3 transcription start sites of the *shot* locus together with the specific sequences recognised by the DSRF transcription factor (squares in magenta) (adapted from *Hahn et al., 2016*). Dorsal and ventral TCs from control (**G and J**) *bs* (**H and K**) mutant embryos. The tracheal overexpression of *Shot* is sufficient to restore the growth of TC subcellular lumina in *bs* mutant background (**I, L**). (**M**) Quantification of TCs with an extended lumen: *bs/ +* (n = 350); *bs/bs* (n = 280) and *bs/bs*;btl >Shot (n = 210). \*\*\* p-value<0001. Statistics by two-tailed Student's *t*-test.

The online version of this article includes the following source data and figure supplement(s) for figure 7:

**Source data 1.** Quantification of blistered mutant phenotypes and rescue by Shot.
**Figure supplement 1.** MTs and actin disorganisation in bs mutant TCs is rescued by Shot Dorsal.

(*Figure 7A,B and E*). To rule out the possibility of DSRF regulating Shot protein levels, we analysed Shot mRNA in control and *bs* embryos by fluorescent in-situ hybridisation. We found that in *bs* TCs Shot mRNA levels were lower than in *bs/+* embryos (*Figure 7C,D*). To confirm that the luminal phenotype observed in *bs* TCs was due to lower Shot levels, we analysed the TC phenotype of *bs* embryos upon tracheal expression of *shot* in these cells. We observed that increasing *shot* expression in TCs resulted in rescue of de novo lumen formation in *bs* TCs (*Figure 7G–M*) and recovery of cytoskeletal organisation (*Figure 7—figure supplement 1*). Taken together these results indicate that at least part of the luminal phenotypes associated with *bs* LOF in TCs are due to lower levels of Shot.

## Shot and tau functionally overlap during subcellular lumen formation and branching

Previous *Drosophila* work suggested that Shot could display potential functional overlap with Tau in microtubule stabilisation (*Alves-Silva et al., 2012*; *Voelzmann et al., 2016*). To assess this functional overlap during TC subcellular branching, we started by overexpressing Tau-GFP in TCs using GAL4 induced expression (*Murray et al., 1998*). Upon overexpression of Tau in otherwise *wt* TCs, we detected ESLs in 93% of TCs, which is comparable to the ShotOE phenotype (*Figure 8A–C*). Like in ShotOE, this effect was dosage dependent, with more TCs with ESLs when more Tau copies were expressed (*Figure 8C*). We then tried to rescue the *shot* LOF phenotype by targeted expression of Tau in TCs. Again, this effect was dosage dependent. We achieved a 64% rescue of the *shot* mutant phenotype with two copies of Tau expressed, indicating that Tau can execute a similar function to Shot in de novo subcellular lumen formation (*Figure 8D,J* and *Figure 8—figure supplements 1* and *2*). We then analysed TCs double mutant for *shot³* and *tau^{MR22}* null alleles (*shot-tau*). These double mutants showed higher numbers of TCs without lumen (85%) than TCs from *shot³* (22%) or *tau^{MR22}* (3%) alone, or a mere sum of these phenotypes, indicating a synergistic genetic effect between *shot* and *tau* (*Figure 8D–H*). These effects were not due to differences in tracheal cell number or fate (*Figure 8—figure supplement 3*). Furthermore, using a mouse Tau antibody, we could detect Tau colocalizing with the growing lumen in TCs (*Figure 8K*). These results indicate that, as seen in neurons (*Voelzmann et al., 2016*), in tracheal TCs Shot and Tau functionally overlap in subcellular lumen formation and branching.

## Shot is required for subcellular luminal branching at larval stages

During larval stages, TCs ramify extensively to form many branches from the same cell body, long cytoplasmic extensions that form one cytoplasmatic membrane-bound lumen each (*Best, 2019*; *Ghabrial et al., 2011*). We questioned if Shot was also necessary for the subcellular branching and lumen extension in these larval cells. To answer this, we expressed different isoforms of Shot, Shot-RNAi and Tau in TCs from embryonic stages with a TC-specific driver (DSRF-GAL4) and analysed the phenotypes on branching and ESL formation at the end of the larval stages (*Figure 9*). Downregulation of Shot induced TCs with lower levels of branching and fewer lumina (*Figure 9B,G,I*). Whereas in control TCs each branch is filled by a subcellular lumen, in Shot-RNAi TCs these were reduced to 37% of the TCs and even so absent in most branches (*Figure 9B and G*). Also, on average, each control TC develops 16.9 ± 1.4 branch points (n = 10), but Shot-RNAi TCs only developed an average of

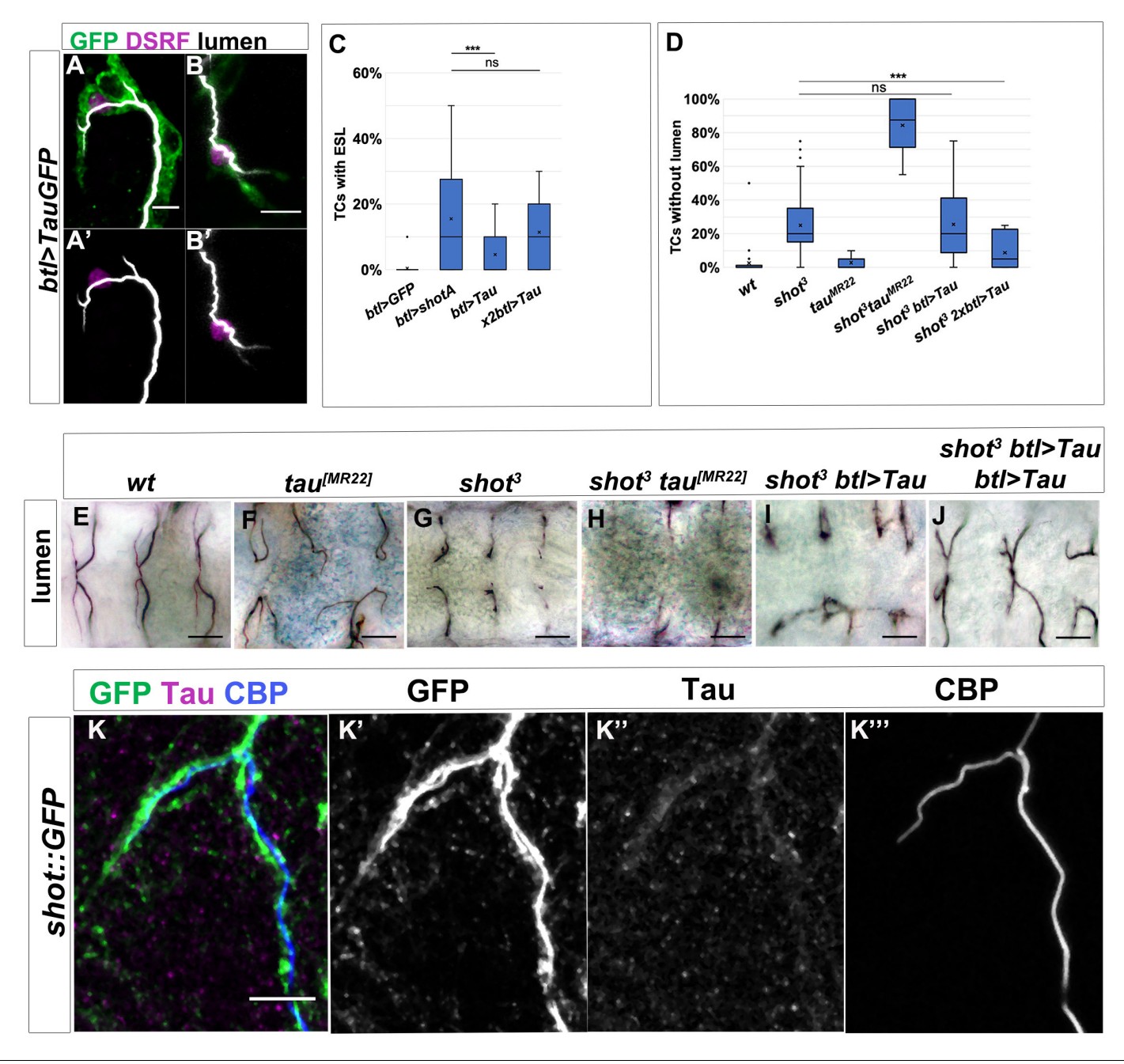

**Figure 8.** Shot and Tau functionally overlap during subcellular lumen formation. (**A–B**) DB (**A**) and GB (**B**) embryonic TCs expressing *tauGFP* in the tracheal system, stained with GFP (green), CBP (white) and DSRF (magenta), showing the ESL phenotype induced by Tau overexpression. In A' in B' lumen and TC nuclei are shown, anterior side on the left, dorsal side is up; scale bar 5 µm. (**C**) Quantification of TCs with ESL in embryos overexpressing GFP (n = 240); *ShotA* (n = 400); one copy of *btl >tauGFP* (n = 440) or two copies of *btl >tauGFP* (n = 300). ***p-value<0.001; ns refers to a p-value>0.1. Statistics by two-tailed Student's *t*-test. (**D**) Quantification of TCs without subcellular lumen in control (n = 820), *shot³*(n = 600), *tau[MR22]* (n = 180), *shot³; tau[MR22]*(n = 180), *shot³; btl >Tau* (n = 440) and *shot³; btl >Tau btl >Tau* (n = 260) embryos. ***p-value<0.001; ns refers to a p-value>0.1. Statistics by two-tailed Student's *t*-test. (**E–J**) Dorsal view of TCs from st. 16 embryos (genotype indicated) stained with anti-Gasp. *tau* deletion mutant does not display a subcellular lumen phenotype (**D and F**) but enhances the effect of *shot* mutation in the double mutant *shot³; tau[MR22]*. One copy of Tau is not sufficient to rescue *shot³* (D and I, n = 400) but two copies rescues the *shot* LOF TC phenotype (D and J n = 260). Scale bars 10 µm. (**K**) Tau is detected in embryonic TCs. Embryonic *shot::GFP* dorsal TC stained with GFP (green in K, grey in K'), anti-Tau antibody (magenta in K, grey in K') and CBP (blue in K grey in K'). Scale bar 5 µm.

The online version of this article includes the following source data and figure supplement(s) for figure 8:

**Source data 1.** Quantification of tau mutant and overrexpression phenotypes and rescues.

*Figure 8 continued on next page*

*Figure 8 continued*

**Figure supplement 1.** MT disorganisation in *shot* mutant TCs is rescued by ShotA and Tau, but not ShotC.
**Figure supplement 2.** Actin disorganisation in *shot* mutant TCs is rescued by ShotA and Tau, but not ShotC.
**Figure supplement 3.** Double *shot³;tau^MR22* mutant embryos display defects in lumen formation but not in tracheal cell number or TC fate.

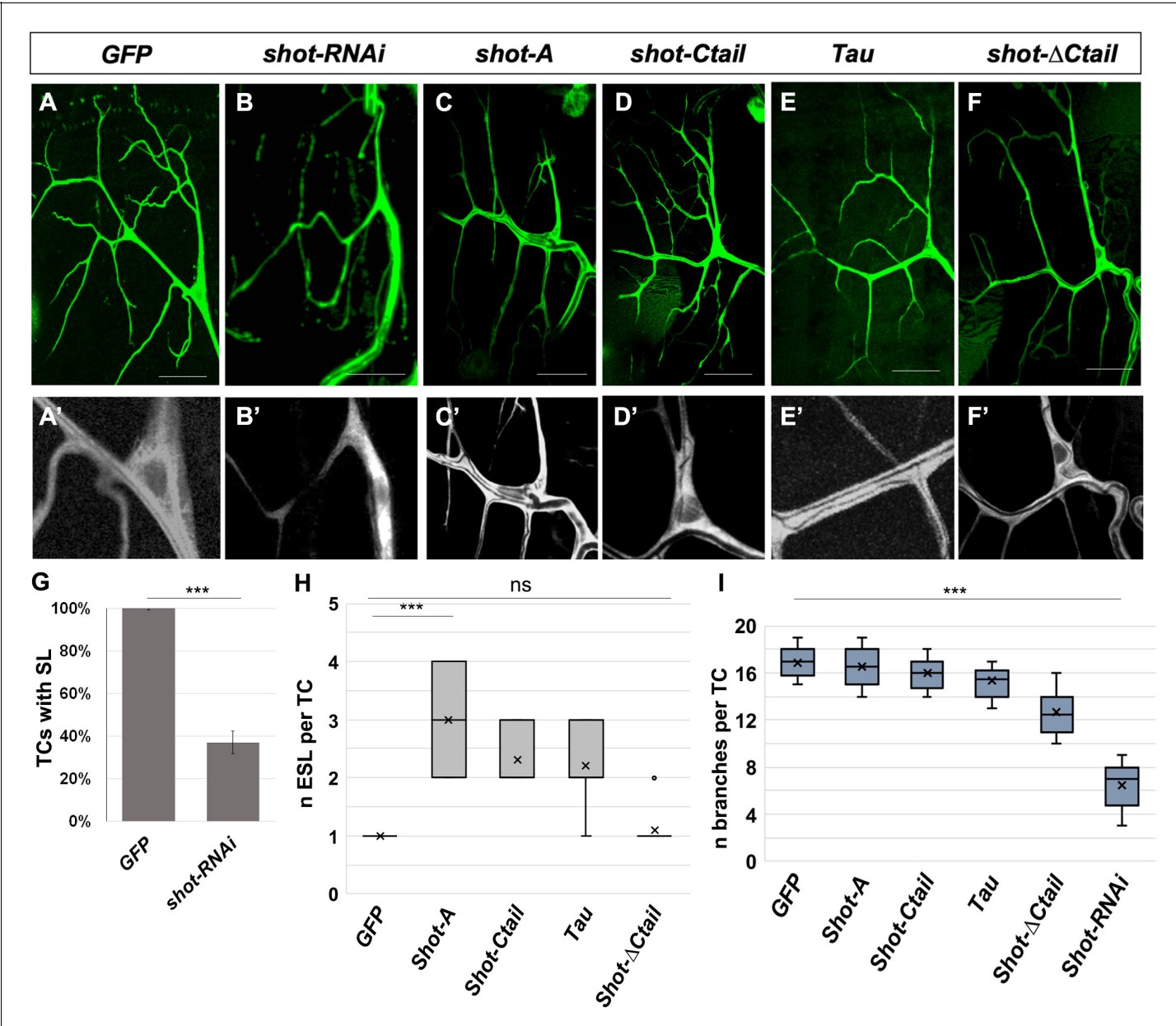

**Figure 9.** Shot and Tau modulate luminal branching in larval TCs. Wandering larval (L3) TCs expressing only GFP (**A**) and different Shot and Tau constructs (**B, C, D, E, F**) under the control of a tracheal DSRFGAL4 driver (all except **A** and **E** where the driver used was btlGAL4). (**A, A'**) UASGFP (n = 8) (**B, B'**) UASshotRNAi, UASGFP (n = 8); (**C, C'**) UASShotA-GFP (n = 10); (**D, D'**) UASshotCtail-GFP (n = 8); (**E, E'**) UASTauGFP (n = 8); (**F,F'**) UASshotΔCtail-GFP (n = 8). Scale bars 50 µm. (**G**) Quantification of the percentage of TCs with subcellular lumen; (**H**) quantification of the number of ESL per TC; (**I**) quantification of the number of branches per larval TC. *** represent a p-value<0.001; ns refers to a p-value>0.1. Statistics by two-tailed Student's *t*-test.

The online version of this article includes the following source data for figure 9:

**Source data 1.** Quantification of larval branching phenotypes.

6.5 ± 0.6 branch points each (n = 8) (*Figure 9B and I*). We then overexpressed the long isoform of Shot (ShotA-GFP aka ShotOE condition) and could not detect extra branching points in TCs, suggesting that more than just an increased Actin-MT crosstalk is needed for the induction of TCs with supernumerary cytoplasmatic extensions (*Figure 9C*). Nonetheless, overexpression of ShotA, ShotC-tail and Tau induced ESL in TCs, with two or more lumina in all TCs analysed (n = 10) (*Figure 9C–E and H*). Like in embryos, targeted expression of Shot-ΔC-tail did not induce ESL in larval TCs (*Figure 9F and H*). Taken together, these results indicate that Shot is necessary for larval lumen formation and branching and that Actin-MT crosstalk by Shot or Tau is sufficient for ESL formation within each TC cytoplasmatic extension.

## Discussion

In this study, we analysed the importance of MT-actin crosstalk through Shot and Tau in subcellular lumen formation in *Drosophila* embryonic and larval tracheal cells. Our work reveals novel insights into the formation of lumina by single-cells. First, that a spectraplakin in involved in the crosstalk between actin and MTs in tracheal TCs and that this crosstalk is necessary for de novo lumen formation. Absence of Shot leads to defects in MT and actin organisation and a profound alteration of the cytoskeleton in TCs (*Figure 10A,C*). Consequently, membrane delivery is disrupted and a novel subcellular lumen cannot be formed. Second, that once a primary lumen is formed de novo in TCs, neither actin-MT crosstalk, nor supernumerary centrosomes, are necessary for the formation of new

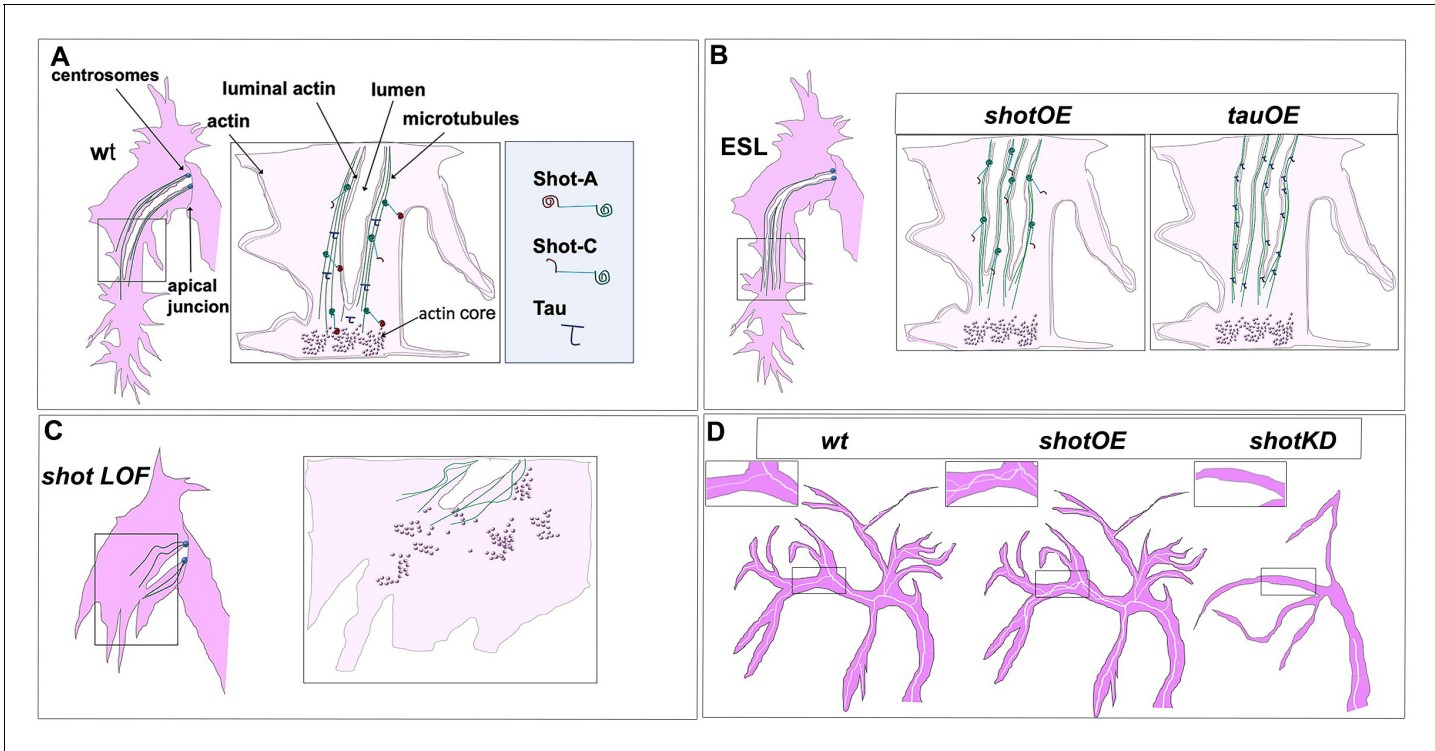

**Figure 10.** Shot and Tau dynamically modulate the cytoskeleton during subcellular lumen formation. Schematic representation of st.16 embryonic (**A, B, C**) and third instar larval (**D**) TCs; cytoplasm is in pink and luminal space in white. (**A**) Cytoskeletal components in a *wt* embryo with the actin-network (dark pink) and MTs (green). Shot and Tau are able to organise the cytoskeleton by crosslinking MTs and actin; Shot (represented with the actin domain in red and the MT-binding domain in green) mediates the crosstalk between actin and MTs as the longer isoform (ShotA), but shorter isoforms lacking part of the ABD were reported not to bind/or very weakly bind actin (ShotC). Tau is represented in blue. (**B**) ESLs are formed by overexpressing *shot* or *tau* by an excess of MT stabilisation from the pre-existing lumen, which probably acts as a MTOC in this case. ESLs can be induced by Shot isoforms with affected (ShotC) or without ABD (ShotC-tail). (**C**) In the absence of the longer isoform of Shot (ShotA) proper cytoskeletal organisation, is not established, by defective MT-actin crosslinking, and cell elongation and lumen formation fail to occur. (**D**) Schematic representation of larval TCs in *wt*, in *shotOE* (or *tauOE*) where ESLs are formed without concomitant single-cell branching and in *shot* KD, where both single-cell and luminal branching are reduced.

supernumerary lumina (ESLs). New lumina can arise from branching points along the length of the pre-existing lumen, only by MT stabilisation by isoforms of Shot lacking entirely the ABD (*Figure 10B,D*). In these cases, we can form ESLs acentrosomally, perhaps from the MTOC activity provided by the gamma-tubulin present along the crescent lumen (*Gervais and Casanova, 2010*) or by other types of MTOCs. Third, spectraplakin activity is necessary to organise MTs and actin in TCs; without Shot TCs exhibit a disrupted MT and actin cytoskeleton, which can be restored by tissue specific expression of this spectraplakin. Fourth, increased levels of Shot are induced in TCs by DSRF, and Shot can rescue the subcellular lumen formation phenotypes in *bs* mutants. This agrees with previous observations in other systems where *bs* and *shot* mutants display similar phenotypes (*Prout et al., 1997*). And fifth, high-levels of Tau can replace Shot in subcellular lumen formation and branching.

## Shot promotes subcellular branching by organizing and mediating the crosstalk between microtubules and actin

Previously, it was shown that Shot was involved in tracheal fusion cell anastomosis during embryonic development (*Lee and Kolodziej, 2002a*). It was observed that Shot accumulates at E-cadherin-dependent contacts between fusion cells and *shot* LOF disrupts this contact leading to cell-fusion phenotypes. In these cells, interactions of Shot with F-actin and microtubules are functionally redundant and both targeted expression of ShotC or ShotA is sufficient to rescue the cell-fusion phenotype (*Lee and Kolodziej, 2002a*). Our results are more akin to what has been reported in neuronal growth cones, and both actin and MT-binding domains of Shot are required for TC extension and subcellular lumen formation (*Figure 10A*). In neurons, like in tracheal cells, ShotC is unable to rescue the phenotype caused by *shot* LOF, which is only rescued by expression of the full-length ShotA isoform (*Lee and Kolodziej, 2002b*). Shot has also been shown to be required for sealing epithelial sheets during dorsal closure (*Takács et al., 2017*). In these epithelial cells, Shot acts as a MT-actin crosslinker to regulate proper formation of the MT network. As in the case of tracheal TCs presented here, the actin- and microtubule-binding activities of Shot are simultaneously required in the same molecule, indicating that like in TCs Shot is engaged as a physical crosslinker also during dorsal closure (*Takács et al., 2017*).

MTs and the actin cytoskeleton perform many functions in tracheal TCs that are regulated by different actin- and MT-binding proteins. While mediators of actin function, such as Ena (*Gervais and Casanova, 2010*), and of MT function, like D-Lissencephaly-1 (DLis-1), have been identified previously, we show here that Shot is able to mediate crosstalk between MTs and actin during subcellular lumen formation. In Shot LOF conditions, MTs and actin are disorganised. Consequently, this Shot crosslinking function is essential for de novo lumen formation and extension. It has been previously described that in TCs of mutants affected in MT organisation, the actin-network is not perturbed (*Gervais and Casanova, 2010*), so the 'actin phenotype' observed in *shot* LOF cannot be a consequence of defects in the MT network. This observation indicates a possible spectraplakin function in organizing TC actin in agreement with previous observations that Shot and ACF7 can promote filopodia formation (*Lee et al., 2007*; *Sanchez-Soriano et al., 2009*).

## Shot expression is regulated by DSRF in TCs

Our results show that molecular levels of Shot are important for cytoskeletal rearrangements, indicating that there is a dosage dependent effect in lumen formation and extension as well as in luminal branching events. Shot is present in many cells during development but Shot level regulation is likely to be more important in cells such as neurons and tracheal terminal cells, due to their morphology (*Voelzmann et al., 2017*). *bs*/DSRF is a TC-specific transcription factor, whose expression is triggered by Bnl signalling (*Guillemin et al., 1996*; *Sutherland et al., 1996*), and is required for TC cytoskeletal organisation (*Gervais and Casanova, 2010*). DSRF has also been shown to be necessary not just for the establishment of TC fate, but to ensure the progression of TC elongation (*Gervais and Casanova, 2011*). Cytoskeletal organisation and remodelling as well as TC elongation are tightly coupled during subcellular lumen formation and in *bs* mutants actin accumulation was impaired at the TC tip (*Gervais and Casanova, 2010*). We observe a similar actin phenotype in Shot mutants (*Figure 5A–D*) suggesting that the actin defects observed in DSRF mutants may be due to a lower expression of Shot in these cells.

## Shot and Tau functionally overlap in subcellular lumen formation and branching

It has been suggested that spectraplakins functionally overlap with structural microtubule-associated-proteins (MAPs). Shot displays a strong functional overlap with Tau in MT stabilisation leading to the adequate delivery of synaptic proteins in *Drosophila* axons (*Voelzmann et al., 2016*). In addition, it has been proposed that a loss of MAP function in mammals results in a relatively mild phenotype due to a functional compensation accomplished by spectraplakins (*Morris et al., 2011*; *Riederer, 2007*). Furthermore, the effect of the complete lack of Shot function during dorsal closure is very subtle (*Takács et al., 2017*), hinting that in another *Drosophila* organ, Shot function might have overlaps with other MAPs.

Our overexpression and genetic data suggest that also in the context of subcellular lumen formation these two proteins functionally overlap. When we tested the tracheal overexpression of *Tau* in *wt* background, we observed extra-subcellular lumina with morphology very similar to the one caused by ShotOE. Moreover, Tau overexpression in tracheal cells was able to rescue the *shot* LOF phenotype similarly to ShotA expression. We propose that Tau's rescuing capability does not depend only on its classical MT-stabilisation activity, since expression of ShotC and ShotC-tail in tracheal cells was not able to restore subcellular lumen formation. Tau MT-binding is probably just one of its functions in TCs. In fact, Tau has been shown to co-organise dynamic MTs and the actin-network in cell-free systems and growth cones (*Biswas and Kalil, 2018*; *Cabrales Fontela et al., 2017*; *Elie et al., 2015*). Our rescue and double mutant analyses suggest that in TCs, Shot and Tau functionally overlap in organizing the coordination between MT-bundling and actin cytoskeleton crosstalk (*Figure 10A,B*).

## Larval lumen formation and branching

TC subcellular lumen formation starts at embryonic stages but most of its elongation and branching occurs during the extensive body growth of the third instar larva (L3). Some mutants have been reported to generate larger TCs with higher numbers of branches. Such mutants included the Hippo pathway member *warts/lats1* (aka *miracle-gro*), and the TOR pathway inhibitor, *Tsc1* (aka *jolly green giant*) (*Ghabrial et al., 2011*). In addition, activation of the FGF Receptor (Btl) pathway in TCs gives rise to ectopic branches (*Jarecki et al., 1999*; *Lee et al., 1996*). Interestingly, in all these cases, mutant TCs develop a higher number of branches but no reported ESL per branch. In larvae, as in embryonic TCs, actin is present at the basal plasma membrane and at the luminal/apical membrane. The connection between the basal actin network and the outer plasma membrane is made through Talin, which links the network to the extracellular matrix (ECM) via the integrin complex (*Levi et al., 2006*). Regulation of the luminal actin is done by Bitesise (Btsz), a Moe interacting protein (*JayaNandanan et al., 2014*). These interactions with actin are required for proper TC morphology, and mutations in either the *Drosophila* Talin gene *rhea* or *btsz* induce multiple convoluted lumina per TC branch (*JayaNandanan et al., 2014*; *Levi et al., 2006*). *rhea* and *btsz* ESLs seem to be misguided within the TC and present a series of U-turns and loops we did not observe in *shot* mutants. Also, mutations in *rhea* and *btsz* do not induce embryonic TC luminal phenotypes, suggesting that despite their interactions with actin, the mechanism of action during subcellular lumen formation and stabilisation is different. They do not seem to interact with MTs and they might have a more structural/less dynamic role in larval subcellular lumen formation. Our results suggest that Shot is able to induce larval ESLs by the same mechanism as in embryos. By modulating a dynamic crosstalk between MTs and actin that induces acentrosomal luminal branching. However, albeit necessary for larval luminal branching excess Shot alone is not sufficient to induce extra branching in TCs. Perhaps ShotOE TCs are able branch their subcellular lumen but lack a specific spatial cue to induce single-cell branching. This cue could be such as the one provided by a hypoxic tissue secreting the FGFR ligand, Bnl, which would allow for the cytoplasmic extensions needed to increase single-cell TC branching.

## Spectraplakins and lumen formation in other organisms

The spectraplakin protein family of cytoskeletal regulators is present throughout the animal kingdom. In the most commonly studied model organisms we find VAB-10 in the worm *Caenorhabditis elegans*, and, in vertebrates, dystonin (also known as Bullous Pemphigoid Antigen 1/BPAG1) and

Microtubule-Actin Crosslinking Fac- tor 1 (MACF1; also known as Actin Crosslinking Family 7/ACF7, Macrophin, Magellan) (*Voelzmann et al., 2017*). They are usually strongly expressed in the nervous system and most of their functions have been unraveled by studying nervous system development and axonal cell biology (*Zhang et al., 2017*). Spectraplakin roles have also been reported in cell-cell adhesion and cell migration (*Röper and Brown, 2003*). Recently, attention has gone into the role of spectraplakins not only during normal cellular processes but also in human disease, from neurode-generation to infection and cancer (*Zhang et al., 2017*). However, not much is known about a role for spectraplakins neither during lumen formation nor during subcellular branching events. Here, we provide evidence for the involvement of the *Drosophila* spectraplakin Shot in subcellular lumen for-mation and luminal branching. Through its actin- and MT- binding domains, Shot is necessary for subcellular lumen formation and branching (*Figure 10*). This function can be functionally replaced by Tau, another microtubule- associated protein which has been shown to be able to crosslink MTs and actin (*Biswas and Kalil, 2018*). A similar crosslink between MTs and actin may in place during verte-brate lumen formation and in other subcellular branching events.

# Materials and methods

## Key resources table

| Reagent type (species) or resource | Designation | Source or reference | Identifiers | Additional information |
|---|---|---|---|---|
| Genetic reagent (*D. melanogaster*) | *shot³* | Bloomington *Drosophila* Stock Center | BSDC:2282 FBst0005141 | *Lee et al., 2000* |
| Genetic reagent (*D. melanogaster*) | *shot^kakP2* | Bloomington *Drosophila* Stock Center | BSDC:29034 FBst0029034 | *Gregory and Brown, 1998* |
| Genetic reagent (*D. melanogaster*) | *shot^ΔEGC* | F. Jankovics | - | *Takács et al., 2017* |
| Genetic reagent (*D. melanogaster*) | *Rca1^G012* | S.J. Araújo | - | *Ricolo et al., 2016* |
| Genetic reagent (*D. melanogaster*) | *tau^[MR22]* | Bloomington *Drosophila* Stock Center | BDSC:9530 FBst0009530 | *Doerflinger et al., 2003* |
| Genetic reagent (*D. melanogaster*) | *bs^0326* | | BSDC:83157 FBst0083157 | *Guillemin et al., 1996* |
| Genetic reagent (*D. melanogaster*) | *btl::moeRFP* | M. Affolter | - | *Ribeiro et al., 2004* |
| Genetic reagent (*D. melanogaster*) | *btl-Gal4* | M. Affolter | - | - |
| Genetic reagent (*D. melanogaster*) | *DSRF4X-Gal4* | A.Ghabrial M. Metzstein | - | - |
| Genetic reagent (*D. melanogaster*) | *UAS-shot L(A) GFP* | Bloomington *Drosophila* Stock Center | BDSC:29044 (FBst0029044) | *Lee and Kolodziej, 2002a* |
| Genetic reagent (*D. melanogaster*) | *UAS-shot L(C)-GFP* | Bloomington Stock Center | BDSC:29042 FBst0029042 | *Lee and Kolodziej, 2002a* |
| Genetic reagent (*D. melanogaster*) | *UAS-shot L(C)-GFP* | Bloomington Stock Center | BDSC:29043 FBst0029043 | *Lee and Kolodziej, 2002a* |
| Genetic reagent (*D. melanogaster*) | *UAS-shot-LA-ΔCtail-GFP* | N. Sanchez-Soriano | - | *Alves-Silva et al., 2012* |
| Genetic reagent (*D. melanogaster*) | *UAS-shot-LA-Ctail-GFP* | N. Sanchez-Soriano | - | *Alves-Silva et al., 2012* |
| Genetic reagent (*D. melanogaster*) | *UAS-TauGFP* | M. Llimargas | - | *Llimargas et al., 2004* |
| Genetic reagent (*D. melanogaster*) | *UAS-bazooka YFP* | J. Casanova | - | *Gervais and Casanova, 2010* |

*Continued on next page*

*Continued*

| Reagent type (species) or resource | Designation | Source or reference | Identifiers | Additional information |
|---|---|---|---|---|
| Genetic reagent (*D. melanogaster*) | UAS-srcGFP | Bloomington Stock Center | BDSC 5432 FBti0013990 | *Kaltschmidt et al., 2000* |
| Genetic reagent (*D. melanogaster*) | *UAS-shot RNAi* | Bloomington Stock Center | BSSC_64041 FBst0064041 | *Perkins et al., 2015* |
| Genetic reagent (*D. melanogaster*) | *UAS-lifeActRFP* | Bloomington Stock Center | BDSC:58715 FBti0164961 | - |
| Genetic reagent (*D. melanogaster*) | *shot::GFP* | J.Pastor-Pareja | - | *Sun et al., 2019* |
| Antibody | mouse anti GASP | DSHB | ID: AB_528492 2A12 | 1:5 |
| Antibody | rat anti DE-cad | DSHB | ID:AB528120 DCAD2 | 1:100 |
| Antibody | guinea Pig anti CP309 | V. Brodu | - | 1:1000 |
| Antibody | rabbit and rat anti DSRF | J. Casanova | - | 1:500 |
| Antibody | goat anti-GFP | Abcam | Catalog # AB6673 | 1:500 |
| Antibody | rabbit anti-GFP | Invitrogen | Catalog # A11122 | 1:500 |
| Antibody | chicken anti-βgal | Abcam | Catalog # AB134435 | 1:500 |
| Antibody | mouse anti-βgal | Promega | Catalog # 23783 | 1:500 |
| Antibody | mouse anti acetylated tubulin | Millipore | Catalog # 3408 | 1:100 |
| Antibody | guinea pig anti Shot | K. Röper | - | 1:1000 |
| Antibody | mouse anti-Tau-1 | Sigma-Aldrich | Catalog # MAB3420 Clone PC1C6 | 1:200 |
| Antibody | mouse anti-Actin | MP Biomedicals | Catalog # 691001 | 1:500 |
| Antibody | anti-dig POD fragments | Roche | Catalog # 11 207 733 910 | 1:1000 |
| Antibody | Goat Anti-Mouse Cy3 (Polyclonal) | Jackson ImmunoResearch | Catalog # 115-165-003 | 1:500 |
| Antibody | Goat anti-Mouse Alexa555 (Polyclonal) | LIFE TECHNOLOGIES/Thermofisher Scientific | Catalog # A-21424 | 1:500 |
| Antibody | Goat anti-mouse Alexa488 (Polyclonal) | LIFE TECHNOLOGIES/Thermofisher Scientific | Catalog # A11029 | 1:500 |
| Antibody | Donkey anti mouse Alexa647 (Polyclonal) | LIFE TECHNOLOGIES/Thermofisher Scientific | Catalog # A31571 | 1:500 |
| Antibody | Goat anti-chicken Alexa555 (Polyclonal) | LIFE TECHNOLOGIES/Thermofisher Scientific | Catalog # A-21437 | 1:500 |
| Antibody | Goat anti-chicken Alexa488 (Polyclonal) | LIFE TECHNOLOGIES/Thermofisher Scientific | Catalog # A-11039 | 1:500 |
| Antibody | Goat anti-chicken Alexa Fluor 647 (Polyclonal) | LIFE TECHNOLOGIES/Thermofisher Scientific | Catalog # A-21449 | 1:500 |
| Antibody | Goat anti-rabbit Alexa 555 (Polyclonal) | LIFE TECHNOLOGIES/Thermofisher Scientific | Catalog # A-21429 | 1:500 |
| Antibody | Goat anti-rabbit Alexa488 (Polyclonal) | LIFE TECHNOLOGIES/Thermofisher Scientific | Catalog # A11008 | 1:500 |

*Continued on next page*

*Continued*

| Reagent type (species) or resource | Designation | Source or reference | Identifiers | Additional information |
|---|---|---|---|---|
| Antibody | goat anti-rabbit Alexa647 (Polyclonal) | LIFE TECHNOLOGIES/Thermofisher Scientific | **Catalog #** A-21244 | 1:500 |
| Antibody | Goat anti-guinea pig Cy2 (Polyclonal) | Jackson ImmunoResearch | **Catalog #** 706-225-148 | 1:500 |
| Antibody | Goat anti-guinea pig 647 (Polyclonal) | LIFE TECHNOLOGIES/Thermofisher Scientific | **Catalog #** A-21244 | 1:500 |
| Antibody | Donkey anti- mouse Alexa 555 (Polyclonal) | LIFE TECHNOLOGIES/Thermofisher Scientific | **Catalog #** A31570 | 1:500 |
| Antibody | Donkey anti- mouse Alexa488 (Polyclonal) | LIFE TECHNOLOGIES/Thermofisher Scientific | **Catalog #** A-21202 | 1:500 |
| Antibody | Donkey anti- mouse Alexa647 (Polyclonal) | LIFE TECHNOLOGIES/Thermofisher Scientific | **Catalog #** A31571 | 1:500 |
| Antibody | Donkey anti- rabbit Alexa555 (Polyclonal) | LIFE TECHNOLOGIES/Thermofisher Scientific | **Catalog #** A31572 | 1:500 |
| Antibody | Donkey anti- rabbit Alexa488 (Polyclonal) | LIFE TECHNOLOGIES/Thermofisher Scientific | **Catalog #** A-21206 | 1:500 |
| Antibody | Donkey anti- rabbit Alexa647 (Polyclonal) | LIFE TECHNOLOGIES/Thermofisher Scientific | **Catalog #**A31573 | 1:500 |
| Antibody | Donkey anti-goat Alexa 555 (Polyclonal) | LIFE TECHNOLOGIES/Thermofisher Scientific | **Catalog #**A21432 | 1:500 |
| Antibody | Donkey anti-goat Alexa 488 (Polyclonal) | LIFE TECHNOLOGIES/Thermofisher Scientific | **Catalog #** A32814 | 1:500 |
| Antibody | Donkey anti-rat Cy3 (Polyclonal) | Jackson ImmunoResearch | **Catalog #**712-165-150 | 1:500 |
| Antibody | Goat Anti-rat Alexa 488 (Polyclonal) | LIFE TECHNOLOGIES/Thermofisher Scientific | **Catalog #** A11006 | 1:500 |
| Antibody | Goat anti-Rat Alexa Fluor647 (Polyclonal) | LIFE TECHNOLOGIES/Thermofisher Scientific | **Catalog #** A-21247 | 1:500 |
| Antibody | Biotinylated goat anti-mouse IgM (Polyclonal) | Thermo Fischer Scientific | **Catalog #**31804 | 1:500 |
| Sequence-based reagent | Shot mRNA probe | This work | - | 1:100 |
| Other | CBP | J. Casanova | - | 1:500 |
| Other | Fluostain | Sigma-Aldrich | FB28 | 1:300 |
| Other | Vectastain-ABC kit | Vector Laboratories | PKU-400 | 1:200 |
| Other | TSA Cy3 | Akoya Bio | NEL744001KT | 1:100 |

## *D. melanogaster* strains and genetics

*shot[3]* (***Lee et al., 2000***), *shot[kakP2]* (***Gregory and Brown, 1998***), *shot[ΔEGC]* (***Takács et al., 2017***), *Rca1[G012]* (***Ricolo et al., 2016***), *tau[MR22]* (***Doerflinger et al., 2003***), *bs[03267]* (***Guillemin et al., 1996***), *btl::moeRFP* (***Ribeiro et al., 2004***), *btl-Gal4* (Shiga Y., 1996), *DSRF4x-Gal4* (gift from A. Ghabrial)

eLife Research article

Cell Biology | Developmental Biology

UAS-shot L(A) and GFP and UAS-shot L(C)-GFP (*Lee and Kolodziej, 2002a*), *UAS-shot-L(A)-ΔCtail-GFP and UAS-shot-L(A)-Ctail-GFP* (*Alves-Silva et al., 2012*), *UAS-TauGFP* (*Murray et al., 1998* and *Llimargas et al., 2004*), *UAS-srcGFP* (*Kaltschmidt et al., 2000*), *UAS-shot-RNAi* (TRiP.HMJ23381, BDSC), *shot::GFP* (Sun., T., 2019), UASlifeActRFP (BDSC), UAS-bazYFP (*Gervais and Casanova, 2010*). Chromosomes were balanced over LacZ or GFP-labelled balancer chromosomes (BDSC). Overexpression and rescue experiments were carried out either with *btl-GAL4* (BDSC) or *DSRF4X-GAL4* (M. Metzstein) at 25˚C.

## Immunohistochemistry, image acquisition, and processing

All stage embryos, collected on agar plates overnight (O/N), were dechorionated with bleach and fixed for 20 min (or 10 min for MT staining) in 4% formaldehyde, PBS (0.1 M NaCl 10 mM phosphate buffer, pH 7.4)/Heptane 1:1. Washes were done with PBT (PBS, 0.1% Tween). Primary antibody incubation was performed in fresh PBT-BSA o/n at 4˚C. Secondary antibody incubation was done in PBT-BSA at room temperature (RT) in the dark for 2 hr.

For DAB histochemistry (used to recognise 2A12/anti-Gasp antibody) after incubation with secondary antibody (mouse IgM biotinylated antibody) embryos were treated with AB solution for 30 min at R/T (Avidin-Biotinylated Horseradish Peroxidase from Vectastain-ABC KIT of Vector Laboratories 1:200 in PBT).

Embryos were incubated with the DAB solution (DAB 0.12% Nickel-Sulphate-Cobalt Chloride, 0.3 % $H_2O_2$) until black colour was achieved, usually 2–5 min.

The primary antibodies used were: mouse anti-Gasp (2A12) 1:5, rat anti-DE-cad (DCAD2) 1:100, from Developmental Studies Hybridoma Bank (DSHB), guinea pig anti-CP309 (from V. Brodu) 1:1000, rabbit and rat anti-DSRF 1:500 (both produced by N. Martín in J. Casanova Lab), goat and rabbit anti-GFP 1:500 (From Roche and Jackson), chicken, rabbit and mouse anti-βgal 1:500 (Cappel, Promega, Abcam), mouse anti acetylated tubulin 1:100 (Millipore), mouse anti-actin 1:500 (MP Biomedicals) guinea pig anti-Shot 1:1000 (K. Röper), mouse anti-Tau-1 1:200 (Sigma Aldrich). Cy3, Cy2, or Cy5 conjugated secondary antibody (Jackson Immuno Research) or Alexa 488, Alexa 647 and Alexa 555 conjugated secondary antibody (Thermo Fischer Scientific) from donkey and/or goat were used 1:500 in PBT 0.5% BSA. Two probes, to label luminal chitin were used: Fluostain 1:200 (FB28, Sigma), and chitin binding protein CBP 1:500 (produced by N. Martín in J. Casanova Lab). Bright field photographs were taken using a Nikon Eclipse 80i microscope with a 20X or 40X objective. Photoshop 21.2.4 and Fiji (ImageJ 2.1.0) were used for measurements, adjustments and to assemble figures. Fluorescence confocal images of fixed embryos where obtained with Leica TCS-SPE system using 20X and 63X (1.40–0.60 oil) objectives (Leica). Fiji (ImageJ 2.1.0) (*Schindelin et al., 2012*) was used for measurements and adjustments. The images shown are, otherwise stated in the text, max-intensity projection of Z-stack section.

Fluorescent in-situ hybridisation (FISH) *shot m*RNA probe was synthesised using a PCR-based technique. The GAS-2 region was selected as target for the probe. The forward (TAATACGACTCAC TATAGGGAGAAATTCGATACATCTGGCTTG) and reverse (ATTTAGGTGACACTATAGAAGAGTC TGTACTTGCCCTCGCC) primers were used. The gene region of interest, flanked by the T7 and Sp6 sequences, was amplified from previously isolated genomic DNA via PCR under standard PCR conditions. After RNA synthesis, the newly synthesised RNA probe was then purified by precipitation, resuspended in hybridisation buffer, and stored at −20˚C.

Freshly fixed embryos were washed and kept at 56˚C in Hybridisation Buffer for 3 hr for pre-hybridisation. In the last 10 min of pre-hybridisation, probes (1:100 in hybridisation buffer) were prepared for hybridisation. The probes were hybridised with the embryos at 56˚C overnight. The next day the embryos were washed and incubated in POD-conjugated anti-Dig (in PBT) for 1 hr. The fluorescent signal was developed by the addition of Cy3 Amplification Reagent (1:100) diluted in TSA Amplification Diluent and incubation at room temperature in the dark for 10 min. Afterwards, the embryos were antibody stained and then mounted in Fluoromount medium and analysed.

## Quantification and statistics

Total number of embryos and TCs quantified (n) are provided in the figure legends. Measurements were imported and treated in Microsoft Excel, where graphics were generated. Error bars in bar graphics and ±in text denote Standard Error of the Mean (SEM).

Box Plot description: within each box, horizontal central line shows the median; box limits indicate the 25th (bottom) and 75th (top) percentiles as determined by Excel software. Whiskers extend vertically 1.5 times the interquartile range, from the 25th and 75th percentiles. The black **x** in the box represents the mean and the black dots denote observations outside the range (outliers). Statistical analyses were performed applying the T-test. Differences were considered significant when $p < 0.05$. In graphics; **$p < 0.01$, ***$p < 0.001$.

### Time-lapse imaging

Dechorionated embryos were immobilised with heptane glue on a coverslip and covered with Oil 10 s Voltalef (VWR). To visualise tracheal Shot in vivo, *btlGAL4UASShotC-GFP* was used in the indicated backgrounds. Actin in tracheal cells was visualised with *btl::moeRFP* or *btlGAL4UASlifeActRFP* where indicated. Imaging was done with a spectral confocal microscope Leica TCS SP5. The images were acquired for the times specified over 50–75 µm from st. 15 embryos; Z-projections and videos were assembled using Fiji (*Schindelin et al., 2012*).

## Acknowledgements

We are grateful to M Llimargas, J Casanova, V Brodu and our lab colleagues for comments on the manuscript. We thank J Casanova and F Serras for the support given throughout this study. We thank M Affolter, M Llimargas, J Casanova, A Prokop, N Sanchez-Soriano, K Roeper, V Brodu, A Ghabrial, F Jankovics, J Pastor-Pareja and the Bloomington *Drosophila* Stock Center (BDSC) for fly stocks and reagents. Thanks also go to L Bardia, A Lladó, N Giakoumakis, S Tosi and J Colombelli from the IRB-ADMF for assistance and advice with confocal microscopy and software; C Stephan-Otto Attolini from the IRB Bioinformatics/Biostatistics Facility; E Fuentes, R Mendez and M Lledós for assistance in some of the experiments. S.J.A. is an IRB Barcelona Alumni and acknowledges the programme for support. D.R. is the recipient of a Juan de la Cierva post-doctoral fellowship from the Spanish *Ministerio de Ciencia, Innovación y Universidades* (FJCI201732443) and was previously funded by an FPU fellowship (FPU12/05765). This work was supported by the Universitat de Barcelona, Generalitat de Catalunya (2017 SGR 1455) and a grant from the Spanish *Ministerio de Ciencia, Innovación y Universidades* (PGC2018-099465-B-I00).

## Additional information

### Funding

| Funder | Grant reference number | Author |
| --- | --- | --- |
| Ministerio de Ciencia, Innovación y Universidades | PGC2018-099465-B-I00 | Delia Ricolo<br>Sofia J Araujo |
| Generalitat de Catalunya | 2017 SGR 1455 | Delia Ricolo<br>Sofia J Araujo |
| Ministerio de Ciencia, Innovación y Universidades | FJCI201732443 | Delia Ricolo |

The funders had no role in study design, data collection and interpretation, or the decision to submit the work for publication.

### Author contributions

Delia Ricolo, Formal analysis, Investigation, Methodology, Writing - review and editing; Sofia J Araujo, Conceptualization, Formal analysis, Supervision, Funding acquisition, Investigation, Methodology, Writing - original draft, Project administration, Writing - review and editing

### Author ORCIDs

Sofia J Araujo (iD) https://orcid.org/0000-0002-4749-8913

### Decision letter and Author response

Decision letter https://doi.org/10.7554/eLife.61111.sa1

Author response https://doi.org/10.7554/eLife.61111.sa2

## Additional files

### Supplementary files

• Supplementary file 1. Supplementary tables 1 and 2 describing DSRF-specific sequences in upstream and downstream regions of the *shot* gene and the DSRF Positional Weight Matrix (PWM). Description of the quantification of phenotypes from *Figure 3*.

• Transparent reporting form

### Data availability

All data generated or analysed during this study are included in the manuscript and supporting files.

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
