## [Decision Letter]

[Editors' note: this paper was reviewed by Review Commons.]

**Acceptance summary:**

This work is significant because it demonstrates the importance of actin-microtubule cross-linking in de novo lumen formation in single cells and reveals interesting molecular details of this process. The authors focus on the developing tracheal system of *Drosophila*, however, the results have implications for different developmental processes and disease pathologies that involve branching morphogenesis.

**Decision letter after peer review:**

[Editors’ note: the authors submitted for reconsideration following the decision after peer review. What follows is the decision letter after the first round of review.]

Thank you for choosing to send your work, "Coordinated crosstalk between microtubules and actin by a spectraplakin regulates lumen formation and branching", for consideration at *eLife*. Your initial submission has been assessed by a Senior Editor in consultation with a member of the Board of Reviewing Editors and three reviewers. Although the work is of interest, we regret to inform you that the findings at this stage are too preliminary for further consideration at *eLife*.

Specifically, the reviewers felt that this manuscript was well-done, characterizing the requirement for the Shot gene product in an interesting experimental setting and adds a new piece of information to this important protein. The findings presented here definitely progress the field of *Drosophila* tracheal development. However, this manuscript does not sufficiently address how Shot links leading-edge protrusions and centrosomes, how it is organized into pre-lumen tract, and how it contributes to further assembly of luminal membrane and directed secretion. Without clues to these fundamental questions, it appears that this paper is most appropriate for expert readers interested in *Drosophila* cell biology and tracheal development but is less well suited for the broad readership of *eLife*.

[Editors’ note: further revisions were suggested prior to acceptance, as described below.]

Thank you for choosing to send your work entitled "Coordinated crosstalk between microtubules and actin by a spectraplakin regulates lumen formation and branching" for consideration at *eLife*. Your letter of appeal has been considered by a Senior Editor and a Reviewing editor, and we are prepared to consider a revised submission with no guarantees of acceptance.

The major concern of the reviewers was that this manuscript does not provide sufficient novel insight into how the tracheal terminal branch is formed, or how Shot coordinates actin filament and microtubule to organize the lumen membrane organization. Therefore, in the revised version of your paper, please make the conceptual advances achieved by your study more clear.

Furthermore, we expect that you will address the comments of the reviewers from Review Commons to the best of your ability and provide a detailed point-by-point response outlining your revisions.

Please also find below the list of concerns that the Reviewing Editor has emphasized, based on the discussion with the three reviewers:

1) The use of bar plots throughout the manuscript is highly undesirable as bar plots can obscure patterns in the data and hide the full spread of the data.

2) Comments on specific figures/staining procedures.

a) It is hard to distinguish the centrosome staining in Figure 3 from what appears to be background puncta/cross-reactivity (in particular B').

b) Similarly, the acetylated tubulin staining in Figure 4C may require attention. It appears that the morphology of the entire microtubule network changes over time. If this is indeed the case it may warrant an explanation.

c) Paraformaldehyde fixation is generally unable to capture dynamic microtubules and given that Shot is an EB1-dependent plus-end tracking protein this fixation protocol is likely missing this population of microtubules (Figure 2). Methanol fixation has been successfully used in fly embryos before (see: Clohisey, SMR, Dzhindzhev NS, Ohkura H (2014) Kank Is an EB1 Interacting Protein that Localises to Muscle-Tendon Attachment Sites in *Drosophila*. PLoS ONE 9(9): e106112; Shuoshuo Wang, Adriana Reuveny, Talila Volk; Nesprin provides elastic properties to muscle nuclei by cooperating with spectraplakin and EB1. J Cell Biol 25 May 2015; 209 (4): 529-538; G.H. Thomas, D.P. Kiehart; Β heavy-spectrin has a restricted tissue and subcellular distribution during *Drosophila* embryogenesis. Development 1994 120: 2039-2050).

d) Figure 7: A more appropriate comparison would be Shot protein level in TC compared to that in the stalk cell (SC) and fusion cell (FC) and how DSRF mutation influences the relative Shot level in TC compared to SC and FC. It seems odd that the example in Figure 7B shows Shot level in TC appearing much lower than in SC. Since Shot is widely expressed in many cell types, presumably DSRF mutation may reduce Shot level to the level of other tracheal cell types. The data could be improved by including TC, FC, and SC in the same confocal view and measuring Shot levels.

Please note that Shot staining in Figure 4A and B appears more concentrated to growing lumen, compared to Figure 7A and B where the lumen is not included in the picture. Perhaps variation in TC shape gives different impressions of Shot localization in various stages and angles of cell images.

---

## [Author Response]

[Editors’ note: the authors resubmitted a revised version of the paper for consideration. What follows is the authors’ response to the first round of review.]

Specifically, the reviewers felt that this manuscript was well-done, characterizing the requirement for the Shot gene product in an interesting experimental setting and adds a new piece of information to this important protein. The findings presented here definitely progress the field of *Drosophila* tracheal development. However, this manuscript does not sufficiently address how Shot links leading-edge protrusions and centrosomes, how it is organized into pre-lumen tract, and how it contributes to further assembly of luminal membrane and directed secretion. Without clues to these fundamental questions, it appears that this paper is most appropriate for expert readers interested in *Drosophila* cell biology and tracheal development but is less well suited for the broad readership of eLife.

Our work on single-cell branching mechanisms is well integrated within the fields of Cell and Developmental Biology, both subject areas in your journal. Specifically, we believe we provide a solid and novel study on cytoskeletal dynamics during lumen formation and branching, important events during vascular development and remodelling. Furthermore, branching abnormalities inflict greatly in development and disease, making these mechanisms interesting to the wide community of life scientists.

Other points offered as a justification for the denial to publish our work are described in the decision letter and we also do not agree with these. Here is a point by point answer to them:

1) "this manuscript does not sufficiently address how Shot links leading-edge protrusions and centrosomes”. We totally disagree with this statement. We show Shot provides the connection between MT bundles stemming from the centrosome and the actin at the tip of the tracheal terminal cell. We show this in vivo (by live imaging), using different genetic conditions and by analysing cells in fixed embryos. We also dissect which Shot domains are necessary for this crosstalk, the MT binding domain and the actin-binding domain. Furthermore, we show that Shot can stabilize MT bundles and create new branching events leading to extra lumina in terminal cells in embryos and larvae.

2) We do not show "how it is organized into pre-lumen tract”. We not sure what is meant by this statement. Shot is organized into prelumen tracts together with MTs as we show repeatedly throughout our reported work.

3) "and how it contributes to further assembly of luminal membrane and directed secretion” We think this is beyond the scope of this work. We know this is very interesting and, of course, we are very interested on how directed secretion is regulated (by Shot or by other molecules), but the whole question is material for another article.

4) Last but not least, judging by the above-mentioned reasons for rejection, it seems this manuscript only reports an involvement of Shot in cytoskeletal dynamics. However, we also show that Shot is transcriptionally regulated by the *Drosophila* SRF transcription factor (blistered), a novel finding with implications in the transcriptional regulation of cytoskeletal dynamics and epithelial adhesion. And we show that Tau can provide an akin MT-actin crosslinking activity during single-cell branching and lumen formation, another novel finding with broad implications in development and ageing.

[Editors’ note: what follows is the authors’ response to the second round of review.]

The major concern of the reviewers was that this manuscript does not provide sufficient novel insight into how the tracheal terminal branch is formed, or how Shot coordinates actin filament and microtubule to organize the lumen membrane organization. Therefore, in the revised version of your paper, please make the conceptual advances achieved by your study more clear.

We provide solid evidence for the involvement of a spectraplakin in lumen formation, a crosslinking event shown for the first time to be important in tubulogenesis. Our work focuses on the cytoskeleton and we show that Shot helps organize the MT and actin during lumen formation. We think directed membrane secretion is the next step in this work, but we also believe this is beyond the scope of this work and a matter for another paper altogether.

However, we have revised the manuscript to make these cytoskeletal advances more clear. In addition, we provide new evidence that membrane delivery is disrupted in *shot^3^* mutants, as expected by the general disruption of the cytoskeleton induce by the lack of MT-actin crosslinking by Shot (Figure 1—figure supplement 1).

Furthermore, we expect that you will address the comments of the reviewers from Review Commons to the best of your ability and provide a detailed point-by-point response outlining your revisions.Please also find below the list of concerns that the Reviewing Editor has emphasized, based on the discussion with the three reviewers:1) The use of bar plots throughout the manuscript is highly undesirable as bar plots can obscure patterns in the data and hide the full spread of the data.We thank you for this comment, and we agree that that is the case. Where possible, we have redone all plots to Box and whisker plots to fully show the full spread of the data. Our conclusions remain the same in all cases.2) Comments on specific figures/staining procedures.a) It is hard to distinguish the centrosome staining in Figure 3 from what appears to be background puncta/cross-reactivity (in particular B').

We use an antibody anti-CP309 to detect centrosomes in Figure 3. Because this is an antibody staining, it detects all centrosomes in all cells in the embryo. Therefore, what can be seen as puncta, is not background but centrosomes in other tissues surrounding the tracheal TCs. Regarding cross-reactivity, we agree that when there is a subcellular lumen sometimes antibodies cross-react with an unidentified antigen at the luminal surface and in Figure 3B this was what was detected (note that Figure 3A is from an embryo in stages before lumen formation). However, we would like to state that this does not interfere with centrosomal detection.

b) Similarly, the acetylated tubulin staining in Figure 4C may require attention. It appears that the morphology of the entire microtubule network changes over time. If this is indeed the case it may warrant an explanation.

We agree the whole TC morphology changes overtime. This is a developing structure and both shape and the cytoskeleton are changing constantly. This is the reason why, in some instances, we provide snapshots of different stages as is the case of Figure 4 C-E which shows the beginning, middle and final stages of cellular extension. In addition, detection is done in wholemount embryos, using antibodies, which detect MTs in tracheal cells and all other cells in the embryo.

c) Paraformaldehyde fixation is generally unable to capture dynamic microtubules and given that Shot is an EB1-dependent plus-end tracking protein this fixation protocol is likely missing this population of microtubules (Figure 2). Methanol fixation has been successfully used in fly embryos before (see: Clohisey, SMR, Dzhindzhev NS, Ohkura H (2014) Kank Is an EB1 Interacting Protein that Localises to Muscle-Tendon Attachment Sites in *Drosophila.* PLoS ONE 9(9): e106112; Shuoshuo Wang, Adriana Reuveny, Talila Volk; Nesprin provides elastic properties to muscle nuclei by cooperating with spectraplakin and EB1. J Cell Biol 25 May 2015; 209 (4): 529-538; G.H. Thomas, D.P. Kiehart; Β heavy-spectrin has a restricted tissue and subcellular distribution during *Drosophila* embryogenesis. Development 1994 120: 2039-2050).

We apologise if there was not enough detail regarding our fixing protocols. We do not fix with paraformaldehyde, but formaldehyde as stated in the Materials and methods. We have tested many different conditions in order to better detect both microtubules and actin in the same cell, within the whole embryo. During our studies on the influence of centrosomes in subcellular lumen formation we tested methanol fixation, cold fixation, boiling fixation and the rapid fixing protocol we use in this work and we could conclude that rapid fixing gives us the best results for all structures we need to detect. Some of these fixation methods are better for MTs, others for centrosomes, others for actin and luminal structures, but to detect them all, according to our data, rapid formaldehyde fixation is the best (“Fluorescent Analysis of *Drosophila* Embryos,” Chapter 9, in *Drosophila* protocols (eds. Sullivan et al.). Cold Spring Harbor Laboratory Press, Cold Spring Harbor, NY, USA, 2000 and Ricolo et al., 2016).

In addition, regarding Shot, as in Figure 2, we also present time-lapse microscopy videos, where Shot can be analysed without any fixation protocol.

d) Figure 7: A more appropriate comparison would be Shot protein level in TC compared to that in the stalk cell (SC) and fusion cell (FC) and how DSRF mutation influences the relative Shot level in TC compared to SC and FC. It seems odd that the example in Figure 7B shows Shot level in TC appearing much lower than in SC. Since Shot is widely expressed in many cell types, presumably DSRF mutation may reduce Shot level to the level of other tracheal cell types. The data could be improved by including TC, FC, and SC in the same confocal view and measuring Shot levels.Please note that Shot staining in Figure 4A and B appears more concentrated to growing lumen, compared to Figure 7A and B where the lumen is not included in the picture. Perhaps variation in TC shape gives different impressions of Shot localization in various stages and angles of cell images.

We have measured Shot protein accumulation in TCs and stalk cells (SCs) and have added these quantifications to the manuscript. According to our results, in a *bs* mutant background Shot levels in SCs are similar, to controls. However, in TCs they are much lower. We conclude and discuss in the manuscript, that Shot levels in TCs need to be higher than in SCs in order to allow the formation of a subcellular lumen. Regulation of Shot expression in SCs is independent of DSRF and most likely regulated by other transcription factors like on other, non-tracheal cells in the embryo. TC fate and subcellular lumen formation are regulated by DSRF and one of its outcomes is the higher expression of Shot in these cells. These protein data are now backed up by our in situ hybridization experiments (requested by Review Commons reviewers), which show that control TCs display higher levels of Shot mRNA compared to *shot* mutant TCs (Figure 7C).

We would also like to emphasize that quantifications are always done taking into account the whole TCs, taking all the confocal Z-sections. Quantification of Shot protein accumulation (as all other quantifications in our manuscript) is done comparing control and *shot* mutant TCs at the same stages of development. In the case of figure / AB, there is no lumen because this is an early stage 15 embryo (when lumen has not yet been formed) in the control and in *bs* mutant. Since *bs* mutants do not develop a subcellular lumen in most TCs, it is more accurate to measure Shot protein levels in stages before lumen formation. Otherwise, these would not be comparable.

Reviewer #1 (Evidence, reproducibility and clarity):This study provides solid evidences showing a role for the spectraplakin Short-stop (Shot) in subcellular lumen formation in the *Drosophila* embryonic and larval trachea. This subcellular morphogenetic process relies on an inward membrane growth that depends on the proper organization of actin and microtubules (MTs) in terminal cells (TCs). Shot depletion leads to a defective or absent lumen while conversely, Shot overexpression promotes excessive branching, independently on the regulation of centrosome numbers previously shown to be important for the regulation of the lumen formation process (Ricolo et al., 2016). Shot is rather important to regulate the organization of the cytoskeleton by crosslinking MTs and actin. Shot expression in TCs is controlled by the *Drosophila* Serum Response Factor (DSRF) transcription factor. Finally Shot functionally overlaps with the MT-stabilizing protein Tau to promote lumen morphogenesis.The figures are clear, and the questions well addressed with carefully designed and controlled experiments. However, I would have few suggestions that will hopefully make some points clearer.Major comments:– Statistical analyses should be added for comparisons of proportions, including Figure 1E, 1L, Figure 2G-I, Figure 6L, Figure 7K, Figure 8C-D and Figure 9G.

We agree with this and have now redone all graphs and revised all quantifications. We have added error bars in all bar graphs, changed many to box plots (to provide a better grasp of the value-ranges) and have provided statistical analysis where appropriate. We have also redone all graphics and phenotype reporting in relation to total terminal cells (rather than embryos or GBs and DBs TCs), because this is a more stringent and comparable way of quantifying all our results.

– It is not always clear what genotype has been used as the "wt" genotype, as in Figure S2 or Figure 3 for example, this should be added to figure legends.

We have now clarified which flies are used as controls in each experiment throughout the paper. We have left *wt* where flies were *wt* and changed all other cases to either the genotype or “control”.

– Live imaging of Shot has been performed with ShotC-GFP, that cannot bind actin. Don't the authors think ShotA-GFP would reflect more accurately Shot endogenous behavior as it interacts both with actin and MTs? It would be better to show this, even if the results shown here tend to be consistent with Shot endogenous localization shown with Shot antibody staining.

We agree and we apologise about this. We have realized there was a mistake in the figure legend of Video 3, which stated the stock we used to do time-lapse was UASShotCGFP when in fact it was ShotAGFP. We would not have drawn the conclusions without analysing live UASShotA embryos. With this new version of the manuscript, we have corrected the figure legend and provide a new video (Video 4) of a time lapse experiment using UASShotAGFP and btl::moeRFP to detect ShotA and actin in live embryos over time.

– It is of course not possible to generate CRISPR mutant flies with mutations in putative DSRF binding sites in a reasonable amount of time, to confirm that Shot transcription is controlled by DSRF. It would thus be nice to reveal shot mRNA expression with in situ hybridization experiments in wt vs. bs embryos. This would confirm that Shot mRNA is downregulated upon DSRF inhibition and rule out a possible indirect effect on Shot protein stability for example.

We believe the presented 3-way approach (in silico, protein quantification and phenotype rescue) is sufficient to show that Shot expression is regulated by DSRF. It is unlikely that we are dealing with protein stability or other issues, because we can rescue the lumen elongation phenotype by solely expressing Shot in TCs. However, we agree that an in situ hybridization experiment provides data to confirm this, and we now provide a conclusive one. Here, it is clear that Shot expression is higher in control TCs and lower in *bs* TCs (Figure 7, C, D).

– In the same figure, it would also be interesting to show what happens to actin and MTs in bs TCs and to which extent their organization is rescued by Shot overexpression.

We have analysed MTs and actin in *bs* TCs as well as in Shot-rescued TCs. We provide these results in Figure 7—figure supplement 1.

– UAS-EB1GFP does not seem to be an appropriate control in Figure 9 (A and B) since it can affect MT dynamics (Vitre, B. et al. EB1 regulates microtubule dynamics and tubulin sheet closure in vitro. Nat. Cell Biol. 10, 415-421 (2008)). Why not simply use an UAS-GFP?

We have not detected any notorious larval TC phenotypes by overexpressing UASEB1GFP in TCs. Their branching is comparable to that in previous studies (for example, Schotenfeld-Roames, et al., 2014) and there were no detectable luminal branching phenotypes. However, we agree it is more correct to analyse cells with a plain GFP and have repeated the controls for this experiment using DSRFGAL4UASGFP. This is now shown in Figure 9A.

– Shot and probably Tau crosslinking activities are important for lumen morphogenesis with a striking increase in the number of embryos without lumen in shot^3^ and shot^3^ tau^MR22^ mutant embryos. The rescue experiments clearly show that Shot binding to both MT and actin is essential for efficient rescue. The same might apply to Tau since it is able to crosslink actin and MTs (Elie et al., 2015). I believe showing actin and MTs organization in these rescue experiments would be necessary.

We have analysed MTs and actin in *tau*-rescued *shot* mutant TCs. We provide these results in 2 new Figures: Figure 8—figure supplement 1 and 2.

Second, the overexpression experiments indicate that Shot is able to induce extra lumen formation even when unable to bind actin as shown with the increase in the number of supernumerary lumina (ESLs) under overexpression of ShotC and ShotCtail to a lesser extent. This phenotype is also observed under Tau overexpression. This suggest that not crosslinking anymore but rather making MTs more stable could be sufficient to promote extra lumen formation in a wt context. Stabilising MTs by treatment with Taxol might thus be sufficient to promote ESL formation. I am fully aware of the difficulty of treating *Drosophila* embryos with drugs, making this experiment hard to do, but I think this dual function of Shot and Tau (crosslinking actin and MTs to promote branching vs. stabilizing MTs leading to excessive branching) should be discussed.

In Figure 2 we show not just that UASShotC is able to induce ESL but also that UAS-ShotCtail containing only the MT binding domain of Shot is enough to induce ESLs in TCs as the reviewer noted, but we also show that UAS-∆Ctail, which has been reported to bind actin but not MTs is not able to induce ESL (Figure 2 H and J).

We agree Taxol treatment would be a nice experiment to do, however we believe we provide enough evidence that MT stability is enough for ESL induction whereas de novo lumen formation requires crosslinking of MTs to actin. As advised, we have discussed better both Shot and Tau dual function in ESL generation and de novo lumen formation.

Reviewer #1 (Significance):The findings shown in this manuscript shed an important light on the way subcellular morphogenesis occurs. It was known that both actin and MTs were required in this process, particularly during the formation of *Drosophila* trachea (Jayanandanan et al., 2019; Gervais and Casanova, 2010). This work provides additional molecular insights into the way branching morphogenesis from a single cell occurs in vivo, clearly demonstrating a requirement for actin-MT crosslinking mediated by Shot and Tau.This could be of great interest in the field of branching morphogenesis and lumen formation, not only in invertebrates but also in vertebrates where such a crosslinking might occur in the vasculature, the lung, the kidney or the mammary gland for example (Ochoa-Espinosa, A. & Affolter, M. Branching Morphogenesis: From Cells to Organs and Back. Cold Spring Harb Perspect Biol 4, a008243-a008243 (2012)).Reviewer #2 (Evidence, reproducibility and clarity):Summary:The development of branched structures with intracellular lumen is widely observed in single cells of circulatory systems. However, the molecular and cellular mechanisms of this complex morphogenesis are largely unknown. In previous study, the authors revealed that centrosome as a microtubule organizing center (MTOC) located at the apical junction contributes subcellular lumen formation in the terminal cells of *Drosophila* tracheal system. The microtubule bundles organized by MTOC are suggested to serve as trafficking mediators and structural stabilizers for the newly elongated lumen.In this manuscript, they focused on a *Drosophila* spectraplakin, Shot, which have been reported to crosslink MT minus-ends to actin network, in the subcellular lumen formation. The paper started by description of lumen elongation defect of the tracheal terminal cells in the shot^3^ null mutant. The overexpression of full-length and series of truncated form of shot exhibited extra-subcellular lumina (ESL) in TCs, suggesting that Shot is required for the lumen formation in dose dependent manner. They next addressed whether Shot overexpression induces ESL through the supernumerary centrosomes as in Rca1 mutant, however the number of centrosomes was not affected. Moreover, the ESL were sprouted distally from the apical junction, suggesting that Shot operate in different way from the Rca1-dependent microtubule organization. To get mechanistic insight of Shot in the luminal formation, they checked localization of the Shot and found it localized with stable MTs around the nascent lumen and with the F-actin at the tip of the cell during the cell elongation and subcellular lumen formation. In shot^3^ mutant, the MT-bundles were no longer localized to apical region and the actin accumulation at the tip of the cell was also reduced. The rescue experiments using several truncated forms of Shot, and well-designed genetic analysis using various shot mutants revealed that both MT binding domain and actin binding domains are needed to develop the lumen. The expression of shot was under the regulation by terminal cell-specific transcription factor bs/DSRF, and the overexpression of shot in bs LOF mutant suppressed its phenotype, indicated that part of the luminal phenotype of bs mutant in terminal cells are due to lower levels of the activity of shot. Finally, they checked whether Tau can compensate the function of shot in the subcellular lumen formation. The lumen elongation defect in shot mutant was suppressed by tau expression, and tau overexpression phenocopied the shot overexpression-induced ESL. Although tau mutant did not show the lumen formation defects, the double mutant of shot and tau exhibited synergistic effect. Shot was also required for subcellular luminal branching at larval stages.Overall, this work highlighted the importance of Shot as a crosslinker between MT and actin that acts in downstream of the FGF signaling-induced bs/DSRF expression for the subcellular lumen formation. An excess of Shot is sufficient for ESL formation from ectopic acentrosomal branching points. Furthermore, the Tau protein can functionally replace Shot in this context.Major comments:The conclusions were basically supported by the set of data presented in this article, but following points need to be clarified.The truncated form ShotC lacks only half of calponin domain that are essential for the actin binding, thus it is still possible to bind actin to some extent. Although the actin binding activity is reported as "very weak" in the cited references, the quantitative analysis has not been done. Thus, the interpretation and claims based on the experiments using ShotC should be reviewed carefully.

We agree with the reviewer and will revise all the text for resubmission in order to make this unambiguous. However, we would like to remark that our claims are not only based on UAS-ShotC but also in the shotkakP2 allele, which does not contain one of the calponin domains and, more clearly, in isoforms such UAS-Shot C-tail which do not have any ABD (in fact, only have the C-tail part of Shot).

Data set in some places seems fragmented. For example, overexpression study of shot constructs (Figure 2) lacks phenotypic comparison of control (btl Gal4 driven control FP) to compare if phenotypes of shot constructs expression are different from control. Different methods of phenotypic quantification are employed. One was counting embryo number with at least one abnormality among 20 TCs of DB or GB, or the other counting every TC for the presence of lumen/branching conditions. The latter is more stringent measure and is more appropriate for the study of single cell morphogenesis.

We totally agree with the reviewer. We have now revised all quantifications and graphs:

1) We have used *btl>GFP* as control to all overexpression experiments in embryos and DSRFGAL4UASGFP in control larvae.

2) We have made the paper uniform regarding quantifications, which are now all done in relation to total TCs and not embryos.

For this reason, many of the graphs, figure legends and quantification values in the manuscript text are now changed.

– Would additional experiments be essential to support the claims of the paper? Request additional experiments only where necessary for the paper as it is, and do not ask authors to open new lines of experimentation.The all videos were using ShotC isoform which lacks half of the actin binding domain. The truncated isoform is not suitable to observe the localization, especially the colocalization with actin. The videos need to be retaken using full-length Shot at the dosage that does not interfere with normal TC development.

Two videos were done with ShotC and one with ShotA (we have submitted a new Video 4, with ShotA).

Some statements on Moesin and Tau localization sound as if the authors studied Shot interaction with nascent Moe and Tau molecules. This is confusing because fragments of Moe and Tau, but not functional full-length proteins, were used.

We have revised the text to make this unambiguous.

Because the transgenic fly is already present, we assume it would be done in 4 weeks. However, it would be influnced under social circumstances whether the lab facilities are able to access or not.The methods provided seem to be sufficient for reproducing the data by competent researchers, and most of the data are solid and the sample numbers are sufficient for the claims. However, the criteria for phenotypic evaluation differs among graphs and figures, that possibly confuse the readers. Standardized measurement methods are desirable.Reviewer #2 (Significance):In blood capillary and insect trachea, the branching process of single vessel cells involves sprouting of cell protrusions, followed by the lumen extension from the main vessels. The lumen formation involves assembly of plasma membrane components inside of the cytoplasm. Since the luminal membrane is associated with protein complexes common to apical cell membrane, lumen formation is believed to involve redirection of apical trafficking of membranes to intracellular sites (Sigurbjörnsdóttir, Mathew and Leptin, 2014). The authors previously demonstrated that centrosome is an important link of pre-existing lumen to de novo lumen formation, leading to the hypothesis that centrosome-derived microtubules organize lumen membrane assembly.In this manuscript, the authors addressed this issue by looking at the function of Shot/Plakin that has both microtubule and actin binding activities. Shot is an ideal candidate for linking actin-rich cell protrusions in the leading edge to centrosome-associated lumen tip. Indeed, the authors clearly showed that shot is required for lumen extension and overexpressed shot protein associates with intracellular tract rich in microtubules and F-actin. Their findings are definitely a progress in the field of *Drosophila* tracheal development. Having said that, how Shot links leading edge protrusions and centrosomes, how it is organized into pre-lumen tract, and how it contributes to further assembly of luminal membrane and directed secretion, are not well understood yet. Without clues to those fundamental questions, I believe this paper is most appropriate for expert readers of *Drosophila* cell biology and tracheal development.Finally, I feel that the paper includes many data sets and some pictures are not easy to grasp essential points, such as three videos showing localization of overexpressed shot-C, RFP-moesin, and Lifeact.Reviewer #3 (Evidence, reproducibility and clarity):Summary:In their manuscript entitled "Coordinated crosstalk between microtubules and actin by a spectraplakin regulates lumen formation and branching" Ricolo and Araujo characterize the requirement for Short Stop (Shot) in the formation of subcellular tubes in tracheal terminal cells.The authors examined embryos homozygous for shot^3^, a presumed null allele of shot. They found an 80% penetrant defect in seamless tube formation or growth. The phenotype resembles that reported for mutations in blistered, which encodes the *Drosophila* SRF ortholog. The authors find that expression of SRF is not blocked by mutations in shot and later find that bs mutants have decreased levels of shot expression and that shot overexpression can partly suppress the bs tube formation defects.The authors then examine whether the requirement for shot is autonomous to the trachea and find that it is, as pan-tracheal shot RNAi replicates the seamless tube defects.The authors find that overexpression of various Shot isoforms results in the formation of ectopic seamless tubes within terminal cells. Using the various transgenic constructs available for shot, the authors show that the overexpression phenotype is dependent upon the interaction between Shot and microtubules and is dose-dependent.Previous work had shown that ectopic terminal cell tubes also can arise due to increased centrosome number; the authors show that centrosome number is not altered in shot mutants.Shot has well characterized actin and microtubule binding functions, and the authors show that Shot localization overlaps both with microtubules and with actin, and that both cytoskeletal elements are aberrant in shot mutant cells. In a series of experiments utilizing various shot mutant backgrounds and shot transgenes, the authors identify requirements for both Shot-cytoskeleton interactions in the formation and branching of seamless tubes in terminal cells.Finally, the authors examine the requirement for Tau in the same processes. Tau and Shot had previously been found to work together in neurons, and this seems to be true in terminal cells as well. Tau overexpression induces ectopic seamless tubes and can partially suppress shot loss of function. Embryos mutant for tau showed seamless tube directionality defects, but not lumen formation or branching. Embryos doubly mutant for tau and shot showed a more severe seamless tube defect than shot mutants alone – an increase in terminal cells with no lumen from 22% to 85%.Authors also examined terminal cells in larval stages usingDSRF-GAL4 to knockdown shot in terminal cells (rather than pan-tracheal knockdown with breathless).The authors conclude from their studies that Shot, through its interactions with microtubules and the actin cytoskeleton coordinate the outgrowth and branching of subcellular tubes. Overlapping function of Tau and possibly other additional MAPs also act in these processes.The work is largely well done, and the conclusions are supported by the data.Reviewer #3 (Significance):The findings will be of interest to a broad cell biology community as they provide a conceptual advance and may help to focus future work on seamless tubulogenesis. The authors do a good job of placing the results in the context of previous studies.